# Evidence of nematic order and nodal superconducting gap along [110] direction in RbFe$_2$As$_2$

Xi Liu[1], Ran Tao[1], Mingqiang Ren[1], Wei Chen[1], Qi Yao[1], Thomas Wolf[2], Yajun Yan[1], Tong Zhang[1,3] & Donglai Feng[1,3]

Unconventional superconductivity often intertwines with various forms of order, such as the nematic order which breaks the rotational symmetry of the lattice. Here we report a scanning tunneling microscopy study on RbFe$_2$As$_2$, a heavily hole-doped Fe-based superconductor (FeSC). We observe significant symmetry breaking in its electronic structure and magnetic vortex which differentiates the $(\pi, \pi)$ and $(\pi, -\pi)$ directions of the unfolded Brillouin zone. It is thus a novel nematic state, distinct from the nematicity of undoped/lightly-doped FeSCs which breaks the $(\pi, 0)/(0, \pi)$ equivalence. Moreover, we observe a clear V-shaped superconducting gap. The gap is suppressed on surface Rb vacancies and step edges, and the suppression is particularly strong at the [110]-oriented edges. This is possibly due to a $d_{x^2-y^2}$ like pairing component with nodes along the [110] directions. Our results thus highlight the intimate connection between nematicity and superconducting pairing in iron-based superconductors.

[1] State Key Laboratory of Surface Physics, Department of Physics, and Advanced Materials Laboratory, Fudan University, 200433 Shanghai, China. [2] Institute for Solid State Physics, Karlsruhe Institute of Technology, D-76021 Karlsruhe, Germany. [3] Collaborative Innovation Center of Advanced Microstructures, 210093 Nanjing, China. These authors contributed equally: Xi Liu, Ran Tao, Mingqiang Ren. Correspondence and requests for materials should be addressed to T.Z. (email: tzhang18@fudan.edu.cn) or to D.F. (email: dlfeng@fudan.edu.cn)

The discovery of FeSCs has opened a new era in the study of unconventional superconductivity[1–4]. Most FeSCs are found to be proximate to a magnetically ordered state and a nematic electronic state that shares similarities with the cuprates[5–13]. In most undoped and lightly-doped FeSCs, the Fe ions are close to $3d^6$ configuration, which favor a stripe-like collinear antiferromagnetic (AFM) order or spin density wave (SDW), with a wave vector $\mathbf{Q} = (\pi, 0)$ or $(0, \pi)$ (an exception is FeTe, which has an bicollinear AFM state with $\mathbf{Q} = (\pi/2, \pi/2)$). A nematic phase, which breaks the equivalence between $\mathbf{a}$ and $\mathbf{b}$ directions in the Fe-plane, develops at the Neel temperature ($T_N$) or slightly above, and approaches the superconducting dome upon doping[5–13]. There has been increasing evidence showing the nematicity is driven by magnetic fluctuations[14,15], nonetheless the orbital-driven scenario is also proposed, especially for FeSe (ref. [16]). The magnetic and/or orbital fluctuations between the nested Fermi surfaces with a vector around $\mathbf{Q}$ may play an essential role in superconductivity and determine the pairing symmetry[3–5]. Therefore, the relation between nematicity, magnetic order and superconductivity has become one of the central themes in FeSCs.

To provide a unified understanding of FeSCs and even cuprates, it is important to examine such a theme in regimes where the configuration of Fe ions deviates significantly from $3d^6$, as the magnetic interactions, electron correlations, and Fermi surface topology will alter drastically[3,5,17]. In theory, strong correlations are expected for the $3d^5$ case to drive the system into a Mott insulating phase[18,19]. It has been suggested that electronic nematicity may occur via doping a Mott insulator[20], as evidenced in underdoped cuprates[21,22]. Meanwhile, the pairing symmetry of FeSC is also predicted to vary with doping[3,4,17]. For the $3d^{5.5}$ configuration, the dominant spin fluctuations are predicted to relocate to $(\pi, \pi)/(\pi, -\pi)$ due to a change in Fermi surface topology, and consequently, $d$-wave pairing is favored[17,23,24]. This configuration has been realized in $A\mathrm{Fe}_2\mathrm{As}_2$ ($A = $ K, Rb, Cs), the most heavily hole-doped FeSCs[25–42]. They have large Sommerfeld coefficients ($\gamma$)[25,26] and mass enhancement[28,34,36], indicative of strong correlations. Recently, a coherence–incoherence crossover[27] and heavy-fermion like behavior[29] were observed in $A\mathrm{Fe}_2\mathrm{As}_2$, suggesting an orbital-selective Mott transition. Heat transport[30,32], magnetic penetration depth[31], NMR[42], and ARPES[35,37] measurements have suggested gap nodes in $A\mathrm{Fe}_2\mathrm{As}_2$ ($A = $ K, Rb); however, whether the nodes are symmetry protected ($d$-wave pairing)[17,23] or accidental (from anisotropic $s$-wave pairing)[43,44] remains hotly contested. In parallel, neutron scattering[40,41] and NMR[42] studies on $\mathrm{KFe}_2\mathrm{As}_2$ did reveal spin fluctuations that deviated from $(\pi, 0)$. It is thus critical to look for possible nematicity with distinct behaviors and its relation with superconductivity.

In this article, we present a milliKelvin scanning tunneling microscopy (STM) study on $\mathrm{RbFe}_2\mathrm{As}_2$ single crystals. Compared to its sister compound $\mathrm{KFe}_2\mathrm{As}_2$, $\mathrm{RbFe}_2\mathrm{As}_2$ has an even larger $\gamma$ value ($\sim$127 mJ/mol·$\mathrm{K}^2$ (ref. [26])) but a lower $T_c$ ($\sim$2.5 K). Remarkably, we observe significant two-fold symmetry in the quasi-particle interference (QPI) and magnetic vortex cores, while the surface atomic lattice remains four-fold symmetric within the experimental resolution. Particularly, this $C_4$–$C_2$ symmetry breaking is along the diagonal direction, 45° off from the Fe–Fe bond or the $(\pi, \pi)$ direction in the unfolded BZ. This suggests that a new type of electronic nematicity and associated fluctuations developed in $\mathrm{RbFe}_2\mathrm{As}_2$, and such diagonal nematicity was found to persist above $T_c$. Moreover, high-energy-resolved tunneling spectra revealed a clear V-shaped superconducting gap, which can be well fitted by a nodal gap function. The gap is suppressed by both surface Rb vacancies (non-magnetic impurities) and near atomic step edges, suggestive of sign-change pairing. Particularly,

the spatial extension of the suppressed-gap region on [110] oriented edges is found to be much wider than the [100] oriented edges, which is likely due to gap nodes in the [110] directions. Finally, we perform surface K dosing on $\mathrm{RbFe}_2\mathrm{As}_2$ and demonstrate that the $(\pi, \pi)$ nematic state can be suppressed by electron doping, while the superconductivity is subsequently enhanced. The possible origin of the diagonal nematicity and its relation to superconducting pairing is discussed.

## Results

### Surface atomic structure and superconducting gap of $\mathrm{RbFe}_2\mathrm{As}_2$.

The experiment is mostly conducted in a millikelvin STM working at $T = 20$ mK (the surface K dosing was conducted in a 4.5 K STM system). The effective electron temperature ($T_{\mathrm{eff}}$) of the former system is calibrated to be 310 mK (see Supplementary Note 1). Sample preparation and more experimental details are described in Methods. $\mathrm{RbFe}_2\mathrm{As}_2$ is stoichiometric with the $\mathrm{ThCr}_2\mathrm{Si}_2$-type structure (see Supplementary Fig. 1a). It is expected to cleave between FeAs layers and results Rb covered surfaces. Figure 1a shows the typical topography of a commonly observed surface (referred as type A surface). It is atomically flat with some basin-like defects. In the defect-free areas, a square lattice with an inter-atomic spacing of 5.4 Å is observed (Fig. 1a inset). This spacing is $\sqrt{2}$ times the in-plane lattice constant of $\mathrm{RbFe}_2\mathrm{As}_2$ ($\mathbf{a_0} = 3.86$ Å). Besides type A surface, we occasionally observed another type of surface region (type B surface), as shown in Fig. 1b. The topography of type B surface is actually similar to type A, and a 5.4-Å square lattice is also observed in its defect-free areas (Fig. 1b inset). However it displays much stronger anisotropy in the electronic states, as we will show below.

To determine the surface atomic structure, higher-resolution STM imaging near the basin defect is shown in Fig. 1c. We find that there is another square lattice inside the basin defect, with a lattice constant equal to $\mathbf{a_0}$ (as marked by black dots). The outside lattice (marked by circles) is rotated 45° and forms a $\sqrt{2} \times \sqrt{2}$ reconstruction with respect to the inside lattice (see Supplementary Note 2 for more details). The surface structure that best explains these observations is sketched in Fig.1d: the surface is Rb terminated with 50% coverage, and the basin areas are Rb vacancies with an exposed FeAs layer. Surface Rb atoms are surrounded by four As atoms underneath, and formed a $\sqrt{2} \times \sqrt{2}$ lattice with the same orientation as the Fe lattice (as denoted by $\mathbf{a}$, $\mathbf{b}$ hereafter). We note in this model the Rb lattice will have two equivalent occupation site that shifted by 1/2 lattice spacing (see Supplementary Fig. 3e). We indeed observe domain structures formed by these two occupations on low-temperature cleaved sample (see Supplementary Fig. 4), and the orientation of the surface Rb lattice is further confirmed by Laue diffraction combined with STM imaging (Supplementary Fig. 5). Moreover, such a surface structure retains four-fold rotational symmetry, at least within the spatial resolution of STM, and is non-polar due to the 50% Rb coverage, which allows STM to access the intrinsic electronic states of $\mathrm{RbFe}_2\mathrm{As}_2$. A similar surface structure was also observed on cleaved $\mathrm{KFe}_2\mathrm{As}_2$ (ref. [39]).

Figure 1e shows typical tunneling spectra on defect-free areas of type A and B surfaces, within a relatively large energy scale ($\pm 200$ mV). On both regions a pronounced conductance peak is observed slightly below $E_F$ (at about from $-2$ to $-3$ meV). The peak is asymmetric with a higher-intensity shoulder at negative energy. This is likely due to a hole-like band with a top just below $E_F$, as evidenced in the QPI measurements below. The enhanced density of states (DOS) near $E_F$ may underlie the large $\gamma$ value of $\mathrm{RbFe}_2\mathrm{As}_2$. A similar conductance peak near $E_F$ was also observed in $\mathrm{KFe}_2\mathrm{As}_2$ (ref. [39]).

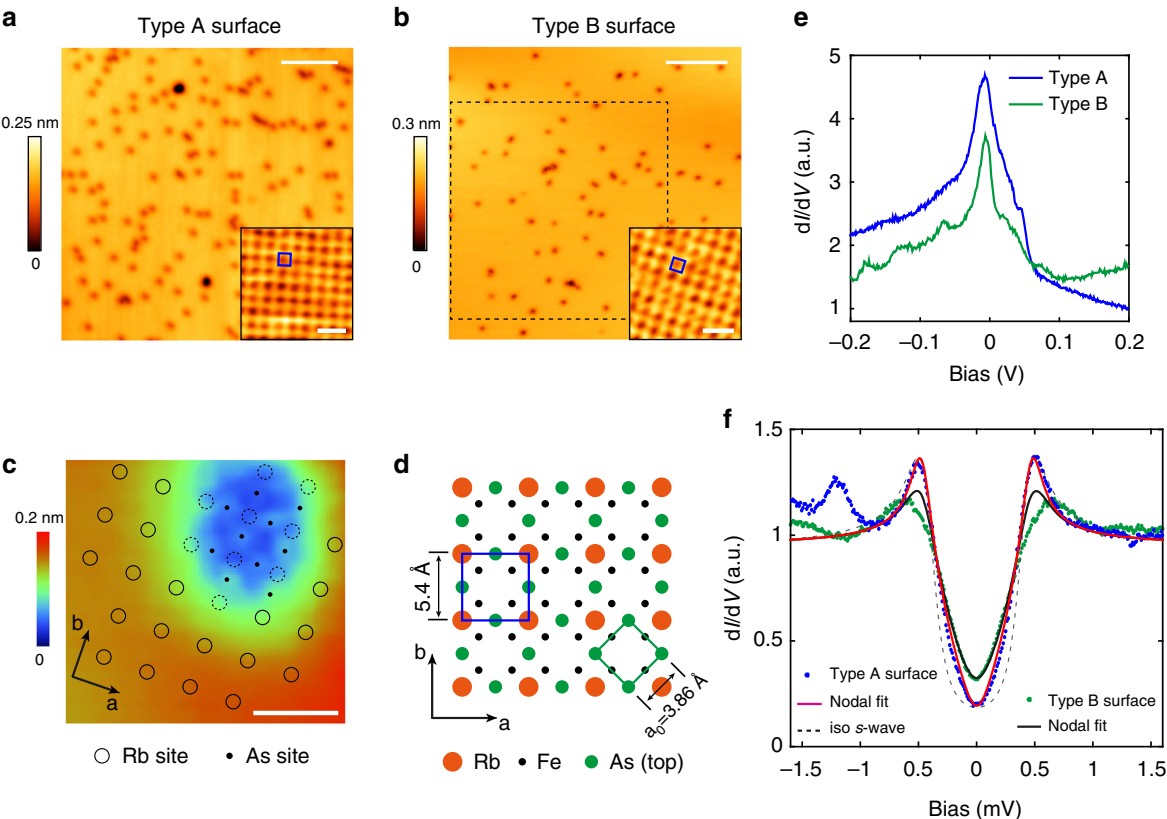

**Fig. 1** Surface atomic structure and tunneling spectrum of cleaved RbFe$_2$As$_2$. **a** Topographic image of type A surface (55 × 55 nm$^2$, $V_b$ = 150 mV, $I$ = 100 pA, scale bar: 10 nm), inset is atomically resolved image of a defect-free area (scale bar: 1 nm), showing a square lattice with a lattice constant of 5.4 Å. **b** Topographic image of type B surface (75 × 75 nm$^2$, $V_b$ = 1 V, $I$ = 10 pA, scale bar: 10 nm). Inset is atomically resolved image of a defect-free area (scale bar: 1 nm), showing a similar lattice to type A surface. QPI mapping on type B surface was performed in the dashed square. **c** A closer image of a large basin defect ($V_b$ = 6 mV, $I$ = 3 nA, scale bar: 1 nm) reveals a different atomic lattice inside the basin. We attribute these atoms to the As layer beneath the surface Rb layer. **d** Sketch of surface atomic structure. The surface is half covered by Rb and forms a $\sqrt{2} \times \sqrt{2}$ (R45°) lattice with respect to the top As layer. **e** d$I$/d$V$ spectra on both surfaces (setpoint: type A surface: $V_b$ = 200 mV, $I$ = 100 pA, $\Delta V$ = 1 mV; type B surface: $V_b$ = 200 mV, $I$ = 50 pA, $\Delta V$ = 1 mV). A DOS peak is observed slightly below $E_F$. **f** Low-energy d$I$/d$V$ spectra taken on type A and B surfaces ($V_b$ = 2 mV, $I$ = 100 pA, $\Delta V$ = 30 μV), and fits to nodal gap and isotropic s-wave gap functions. All the data shown in this figure are taken at $T$ = 20 mK ($T_{eff}$ = 310 mK)

To explore the superconducting state, low-energy tunneling spectra (±1.6 mV) are measured at zero magnetic field. As shown in Fig. 1f, a well-defined V-shaped superconducting gap is observed on type A surface (blue dots), which is the commonly observed case; for type B surface the gap is noticeably broader (green dots). The hump-like structure at negative bias is due to the aforementioned strong DOS peak. The gap on both surfaces are spatially uniform (see Supplementary Fig. 6). We find that the superconducting gap of type A surface can be well fitted with a nodal gap function. The red curve in Fig. 1f is a d-wave fit by using the Dynes formula[45] for the superconducting DOS:

$$N(E)_k = |\text{Re}[(E - i\Gamma)/\sqrt{(E - i\Gamma)^2 - \Delta_k^2}]|, \text{ and } \Delta_k = \Delta_0\cos(2\theta_k).$$

(Note that the gap function $\Delta_k = \Delta_0\cos(4\theta_k)$ which corresponds to nodal s-wave pairing[43] will result in exactly the same DOS). The tunneling conductance is given by d$I$/d$V \propto N(E)_k f'(E + eV)\text{d}k\text{d}E$, where $f(E)$ is the Fermi–Dirac function at $T_{eff}$ = 310 mK. The fitting yields $\Delta_0$ = 0.47 meV and a small $\Gamma$ of 0.03 meV that accounts for additional non-thermal broadening (e.g. impurity scattering). The ratio $2\Delta_0/k_B T_c$ = 4.36. For comparison, anisotropic s-wave gap fit is also plotted in Fig. 1f (dashed curve). It does not match the tunneling spectrum,

especially around the gap bottom. On type B surface, the nodal fit with a similar gap size of $\Delta_0$ = 0.46 meV can match the gap bottom but deviates near the coherence peaks (black curve). The fitting yields a larger $\Gamma$ of 0.09 meV, which may indicate a detrimental effect to superconductivity on type B surface.

**C$_4$ symmetry breaking in QPI and magnet vortex mapping.** Next we turn to examine the electronic structure of RbFe$_2$As$_2$ by performing d$I$/d$V$ mapping. Figure 2a, b are two representative d$I$/d$V$ maps taken on the type A surface in Fig. 1a (at $T$ = 20 mK, $T_{eff}$ = 310 mK), where clear interference patterns can be observed. Figure 2e, f display the raw and symmetrized fast-Fourier transform (FFT) maps at different energies, respectively, which give the **q**-space scattering patterns (see Supplementary Fig. 7 for a complete set of QPI data and raw FFTs). The FFTs display a ring-like structure at relatively high energies ($E$ > 7 meV). However, it becomes diamond-shaped and obviously two-fold symmetric as approaching $E_F$. To identify the orientation of the scattering pattern, in Fig. 2i we plot the FFT map at $E$ = 2.2 meV together with the unfolded BZ (derived from atomic resolved topography and the lattice structure shown in Fig. 1d). It is seen that the two-fold-symmetric axes are along the $(\pi, \pi)$ or $(\pi, -\pi)$ directions (the diagonal of the Fe plaquette). Such C$_4$ symmetry breaking between $(\pi, \pi)$ and $(\pi, -\pi)$ in QPI has never been reported for

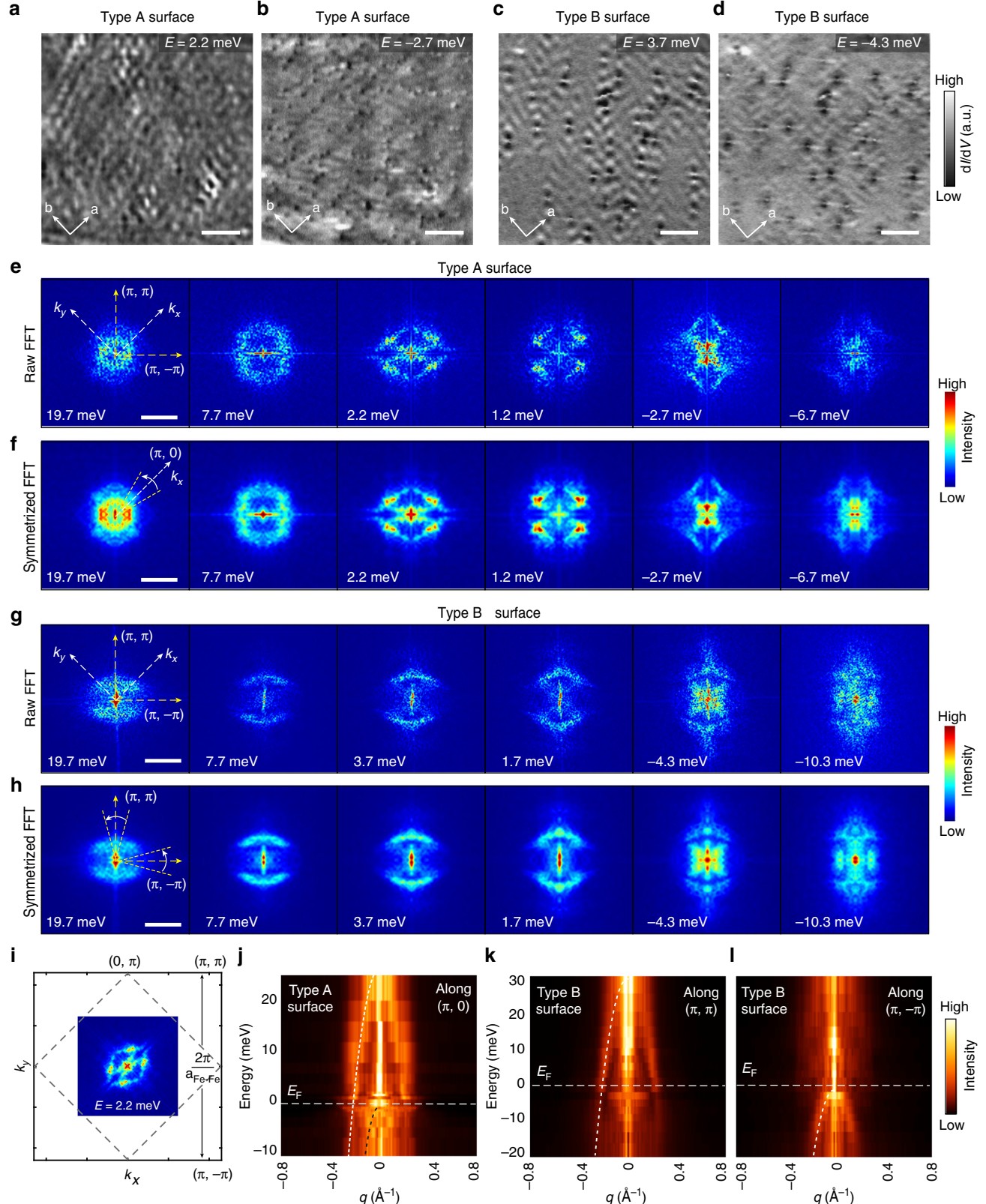

other iron pnictides. Another notable feature is the elongated direction of the scattering pattern rotates 90° as the energy is lowered below $E_F$ (e.g., compare the FFTs of $E = 2.2$ meV and $-2.7$ meV in Fig. 2f, and see also Supplementary Fig. 7). Despite such a complicated evolution, the overall anisotropic scattering patterns changes with energy (see also Supplementary Fig. 7),

indicating that they originate from QPI of anisotropic band(s). To demonstrate the evolvement of such band(s), in Fig. 2j we summarize the FFT profile near $(\pi, 0)$ direction (where strong scattering weight distributed around) at various energy. An overall hole-like dispersion can be seen, and a parabolic fit yields a band top $(E_b) = 27$ meV and Fermi crossing at $q_F = 0.21$ Å$^{-1}$.

**Fig. 2** QPI measurements on RbFe$_2$As$_2$. **a**, **b** Representative d$I$/d$V$ maps taken on type A surfaces. The mapping area is the same as shown Fig. 1a. **c**, **d** Representative d$I$/d$V$ maps taken on type B surfaces. The mapping area is marked in Fig. 1b. (Scale bars in panels **a**–**d**: 10 nm) **e**, **f** Representative raw and symmetrized FFT images of the d$I$/d$V$ maps taken on type A surfaces, respectively (scale bars: 0.3 Å$^{-1}$). **g**, **h** Representative raw and symmetrized FFT images of the d$I$/d$V$ maps taken on type B surfaces, respectively (scale bars: 0.3 Å$^{-1}$). The orientation of the unfolded BZ is marked on the first FFT image of panels **e** and **g**. **i** Sketch of the unfolded Brillouin zone of RbFe$_2$As$_2$ and its relation to the FFT pattern. The C$_4$ symmetry breaking makes ($\pi$, $\pi$) and ($\pi$, $-\pi$) inequivalent. **j** FFT profiles along ($\pi$, 0) direction of type A surface (averaged over a 30° angle that indicated in panel **f**), in which a hole-like dispersion can be observed. Parabolic fit (white dashed curve) gives $E_b$ = 27 meV and $q_F$ = 0.21 Å$^{-1}$. Another hole-like dispersion (black dashed curve) slightly below $E_F$ is observed. **k** FFT profile of type B surface along the ($\pi$, $\pi$) direction (averaged over a 30° angle indicated in panel **h**). **l** FFT profile of type B surface along the ($\pi$, $-\pi$) direction (averaged over a 30° angle), another hole-like band can be seen below $E_F$. Note: All QPI data are taken at $T$ = 20 mK ($T_{eff}$ = 310 mK). The symmetrized FFTs are mirror symmetrized along ($\pi$, $\pi$) and ($\pi$, $-\pi$) directions (see Supplementary Note 3 for more details). Each d$I$/d$V$ maps are taken at a $V_b$ equal to the mapping energy (labeled) and $I$ = 100 pA; the lock-in modulation ($\Delta V$) for each map has an amplitude of 5% $V_b$

More specific interpretation to such QPI will require detailed knowledge on the origin of anisotropic band structure.

Besides commonly observed type A surfaces, on the occasionally observed type B surfaces we find even greater anisotropy. Figure 2c, d g–h show representative d$I$/d$V$ maps and FFTs taken on a 50 × 50 nm$^2$ area marked in Fig. 1b (a complete set of QPI data is shown in Supplementary Fig. 8). Highly anisotropic interference patterns can be seen in the vicinity of surface defects, which are more pronounced along one of diagonal direction of the Fe lattice. The corresponding FFTs now display two arc-like features, which are also along the ($\pi$, $\pi$) in the unfolded BZ, manifesting a strong C$_4$ symmetry breaking. Such arc-like features exist in a relatively wide energy range from –10 to 20 meV (see Supplementary Fig. 8) and disperse with energy as well. In Fig. 2k we show the FFT profile of type B surfaces around the ($\pi$, $\pi$) direction. A hole-like dispersion is observed with $E_b$ = 30 meV and $q_F$ = 0.22 Å$^{-1}$, which are close to the values for type A surface. Thus the basic band structure of type B surface is likely similar to that of type A surface; however, it is driven to be more anisotropic for some reason as discussed later. Despite the anisotropy, the hole-like dispersion in QPI appears to be consistent with a recent ARPES study on RbFe$_2$As$_2$[38], in which a single hole pocket is observed at $\Gamma$. We note that ARPES studies on the sister compound KFe$_2$As$_2$ found three hole pockets at $\Gamma$, which is reproduced in DFT calculations on its paramagnetic state[34–36]. As discussed in ref. [38], this difference could be due to the larger spacing between FeAs layers in RbFe$_2$As$_2$, which enhances the two-dimensionality of the system.

Another notable feature in Fig. 2j is that besides the main hole-like dispersion, there is likely a second band with the top very close to $E_F$ (tracked by black dashed line). Such feature is more clearly seen in the FFT profiles of type B surface along the ($\pi$, $-\pi$) direction, as shown in Fig. 2l. However, it is totally absent in the ($\pi$, $\pi$) direction (Fig. 2k), which also reflects the C$_4$–C$_2$ symmetry breaking. This band may closely relate to the DOS peak observed in d$I$/d$V$ (Fig. 1e), as there could be van-Hove singularity near the band top[39]. Its exact origin is unclear at this stage.

To further investigate this novel C$_4$ symmetry breaking and its effect on the superconductivity, we studied magnetic vortices induced by an external field. Figure 3a shows a zero-bias conductance (ZBC) mapping taken on a 150 × 150 nm$^2$ area of type A surface, under a perpendicular field of $B$ = 0.5 T. A vortex lattice is reflected by the high conductance regions. One sees that the vortex cores display anisotropic shape. To show this more clearly, a spatially averaged core (of 6 vortices) is shown in the Fig. 3c inset. It is slightly elongated along the diagonal of the Fe lattice, consistent with the two-fold symmetry of QPI. At the core center, a zero-bias peak is observed in d$I$/d$V$ (Fig. 3b), and the peak "splits" on moving away from the center. This is typical behavior of vortex core states for a clean superconductor[46]. The core states decay spatially on approximately the scale of the superconducting coherence length ($\xi$). Exponential fits to the

profile along the long and short axes of the vortex core yield $\xi_A^L$ = 15.0(±0.35) nm and $\xi_A^S$ = 12.5(±0.2) nm (Fig. 3c), respectively. For type B surface, ZBC map of a 225 × 225 nm$^2$ area under $B$ = 0.5 T is shown in Fig. 3d. The vortex cores are clearly more anisotropic (a zoom-in of single core is shown in Fig. 3f); while a zero-bias peak is also observed in the core center (Fig. 3e). The coherence lengths for the long and short axes are found to be $\xi_B^L$ = 26.7(±0.7) nm and $\xi_B^S$ = 16.3(±0.5) nm, respectively. As expected, the ratio $\xi^L/\xi^S$ of type B surface (1.63) is larger than that of type A surface (1.2), reflecting stronger anisotropy in the former. Since the spatial shape of the vortex core is intimately related to the underlying band structure[47], the elongated vortex cores provide further evidence for C$_4$ symmetry breaking in the electron states. To see the orientation of the anisotropic core with respect to the **k**-space band structure, we superpose FFT maps (taken near $E_F$) onto the vortex maps as insets in Figs. 3a, d. On type B surface, the vortex is elongated along the direction where the FFT displays a (dispersive) arc-like feature. This is apparently consistent with the BCS expectation that $\xi$ is longer in the direction with larger Fermi velocity ($\xi \sim h v_F/\pi\Delta$). On type A surface, a similar tendency is observed, despite weaker anisotropy in the vortex core and QPI. We note the **k**-space structure of $\Delta$ and the possible nematic order as discussed below should also related to the anisotropy of vortex core[48,49].

So far, the C$_4$ symmetry breaking between ($\pi$, $\pi$)/($\pi$, $-\pi$) has been clearly evidenced in QPI and vortex measurements. The emergence of significant symmetry breaking in such heavily hole-doped region is surprising. It cannot be a surface effect since the surface atomic structure remains four-fold symmetric (Fig. 1d), and there is no bulk structural transition reported for $A$Fe$_2$As$_2$ ($A$ = Rb, K, Cs). Furthermore, the surface Rb vacancies are point-like without noticeable uniaxial anisotropy that may introduce anisotropic QPI, and the shape of the vortex core is also unrelated to surface defects. Thus, the observed symmetry breaking is reminiscent of a nematic-like electron state. Previously, anisotropic QPI, which breaks the symmetry between ($\pi$, 0) and (0, $\pi$), was observed in undoped and lightly-doped iron pnictides, such as Ca(Fe$_{1-x}$Co$_x$)$_2$As$_2$[7], NaFeAs[12], and LaOFeAs[9]. It was commonly considered to be a signature of electronic nematicity with an origin closely related to spin and/or orbital degrees of freedom. Theoretical works have shown that the ($\pi$, 0) stripe AFM (SDW) order tends to open a partial gap along the antiferromagnetic direction, which distorts the Fermi surface and thus the QPI to be two-fold symmetric[50,51]. Such symmetry breaking may even persist above $T_N$ due to short-range spin fluctuations[52]. Here, it is the first time to visualize a C$_4$ symmetry breaking in heavily hole-doped FeSC, and in a 45° rotated direction, this may suggest RbFe$_2$As$_2$ is likely proximate to a stripe-type AFM order or SDW with a **Q** along ($\pi$, $\pi$) direction, which breaks the ($\pi$, $\pi$)/($\pi$, $-\pi$) equivalence. We note that two-fold anisotropic QPI along ($\pi$, $\pi$) has been reported in FeTe films, which exhibit a bicollinear AFM with **Q** = ($\pi$/2, $\pi$/2)[53].

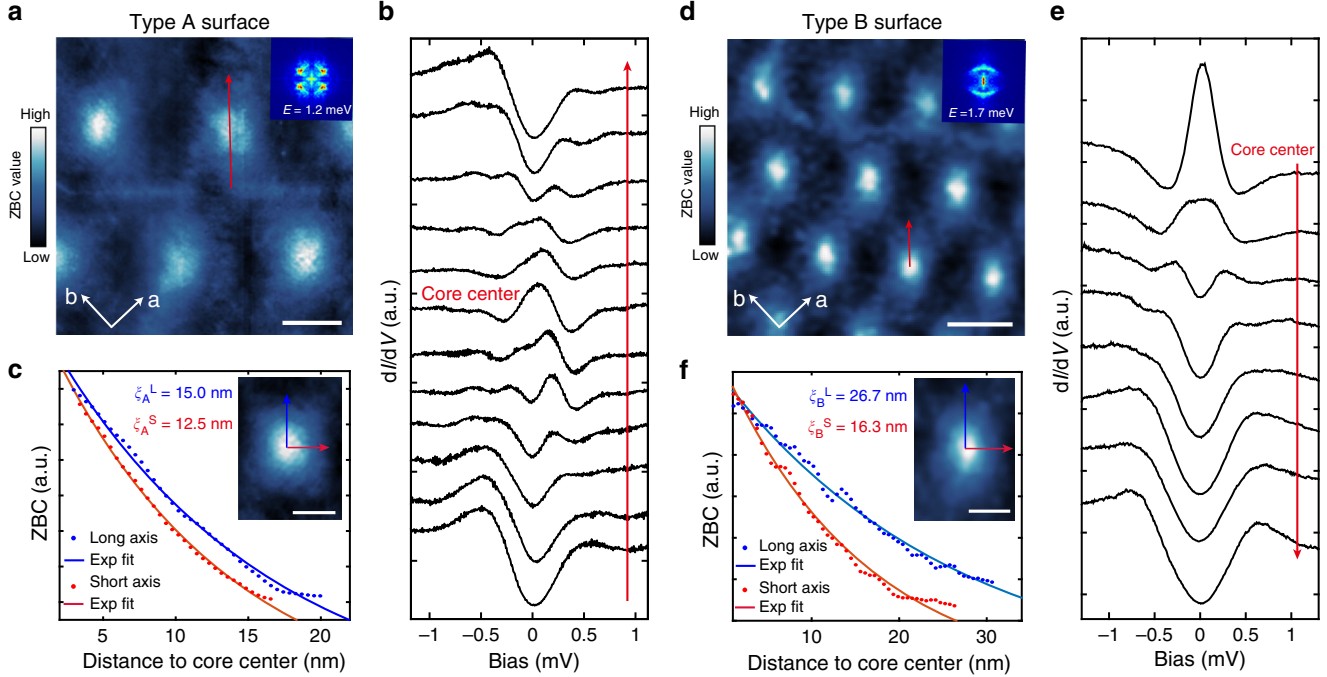

**Fig. 3** Magnetic vortex mapping on RbFe$_2$As$_2$. **a** Zero-bias conductance (ZBC) mapping on type A surface under $B = 0.5$ T (size: $150 \times 150$ nm$^2$; $V_b = 1.2$ mV, $I = 150$ pA, $\Delta V = 50$ µV; scale bar: 30 nm). A spatially averaged core is shown in panel **c**. Inset is an FFT image aligned to the ZBC map. **b** Evolution of the d$I$/d$V$ spectra taken across the vortex core ($V_b = 1.5$ mV, $I = 100$ pA, $\Delta V = 50$ µV), along the red arrow in **a**. A zero-bias peak is observed at the center and splits when leaving the core. **c** ZBC line profiles along the long and short axes of an averaged vortex core (inset image, scale bar: 20 nm). Solid curves are exponential fits, which yield the coherence lengths $\xi^L$ and $\xi^S$. **d** ZBC mapping of type B surface under $B = 0.5$ T (size: $225 \times 225$ nm$^2$; $V_b = 1.5$ mV, $I = 120$ pA, $\Delta V = 50$ µV; scale bar: 50 nm). The elongated direction of the vortex cores is where the QPI shows arc-like features (inset). **e** d$I$/d$V$ spectra taken across the vortex core ($V_b = 1.5$ mV, $I = 100$pA, $\Delta V = 50$µV), along the red arrow in **d**. A zero-bias peak is also observed. **f** ZBC line profiles along the long and short axes of a vortex core on type B surface (inset image, scale bar: 20 nm) and their exponential fits. The difference between $\xi^L$ and $\xi^S$ on type B surface is more significant than on type A surface. (All the data shown in this figure are taken at $T = 20$ mK ($T_{eff} = 310$ mK))

While direct measurements of spin fluctuations in RbFe$_2$As$_2$ are still lacking, spin fluctuations that deviated from $(\pi, 0)$ has been observed in KFe$_2$As$_2$[40–42]. However, whether these fluctuations can drive the $C_4$ symmetry-breaking needs further study. On the other hand, orbital order can also drive nematicity, as proposed for FeSe[16], which will result in an anisotropic Fermi surface and QPI[8]. A similar orbital order was also observed in FeSe$_x$Te$_{1-x}$[54]. No matter which mechanism may apply, details of the anisotropic band structure will depend on material parameters such as Hund's coupling and on-site Coulomb interaction[50,51], which require further investigations.

Assuming the symmetry breaking is from electronic nematicity, there remains a question as to why type B surface shows stronger anisotropy than type A surface, despite their seemingly identical surface lattice structure. We note that for undoped iron pnictides, the $(\pi, 0)/(0, \pi)$ nematicity (and the stripe AFM order) can be enhanced by applying uniaxial pressure[55]. Thus we speculate that type B surface may have local strain (e.g. due to different shrinkage of the sample and its glue upon cooling), which enhances the nematic state in these regions, while even type A surface may also have strain but is likely weaker than type B surface. Nevertheless, even if the strain plays a role here, our observations still imply that RbFe$_2$As$_2$ has a strong tendency or "susceptibility" to form nematic states along $(\pi, \pi)/(\pi, -\pi)$ rather than $(\pi, 0)/(0, \pi)$. Furthermore, it is clear that the superconducting gap features observed on type B surface are significantly broader than on type A surface (Fig. 1f). This reflects a competition between superconductivity and the degree of anisotropy, resembling the anti-correlation between superconductivity and the $(\pi, 0)/(0, \pi)$ nematicity observed in NaFe$_{1-x}$Co$_x$As[13] and

FeSe$_x$Te$_{1-x}$[54]. More evidence on such anti-correlation behavior is shown in the surface K dosing measurement below.

**Effect of Rb vacancy and atomic step edges on superconductivity.** The $(\pi, \pi)/(\pi, -\pi)$ symmetry breaking may have a more profound relation to the superconductivity in RbFe$_2$As$_2$— the interactions and fluctuations of the nematic state could also underlie the electron pairing. Investigating the pairing symmetry of the system can provide further insight on this. In Fig. 1f, the well-defined V-shaped superconducting gap has suggested a nodal pairing, to further investigate the pairing symmetry, we studied the impurity effect that was induced by surface Rb vacancies and atomic step edges. Figure 4a shows an STM image around a Rb vacancy ($V_{Rb}$) on type A surface, while Fig. 4b displays the tunneling spectra taken near it. The superconducting gap has an increased DOS near $E_F$ at the $V_{Rb}$ site, which evidences a local suppression of superconductivity. This gap suppression quickly disappears on moving away from $V_{Rb}$ (the Fig. 4b inset details the gap bottom). An exponential fit to the ZBC value as a function of distance yields a decay length of 1.41 nm (Fig. 4c). Similar gap suppression was also observed near the $V_{Rb}$ on type B surfaces, as shown in Fig. 4d–f. Since Rb vacancies are expected to be non-magnetic, this suppression of the superconductivity suggests a sign-changing pairing[56]. However, we note both $d$-wave and extended $s$-wave pairing with accidental nodes may have a sign-change, as recently suggested in the sister compound KFe$_2$As$_2$ (refs. [35,43,44],). A further way to gain **k**-space information on the gap is to detect its response to sample boundaries. It was predicated that for a nodal pairing

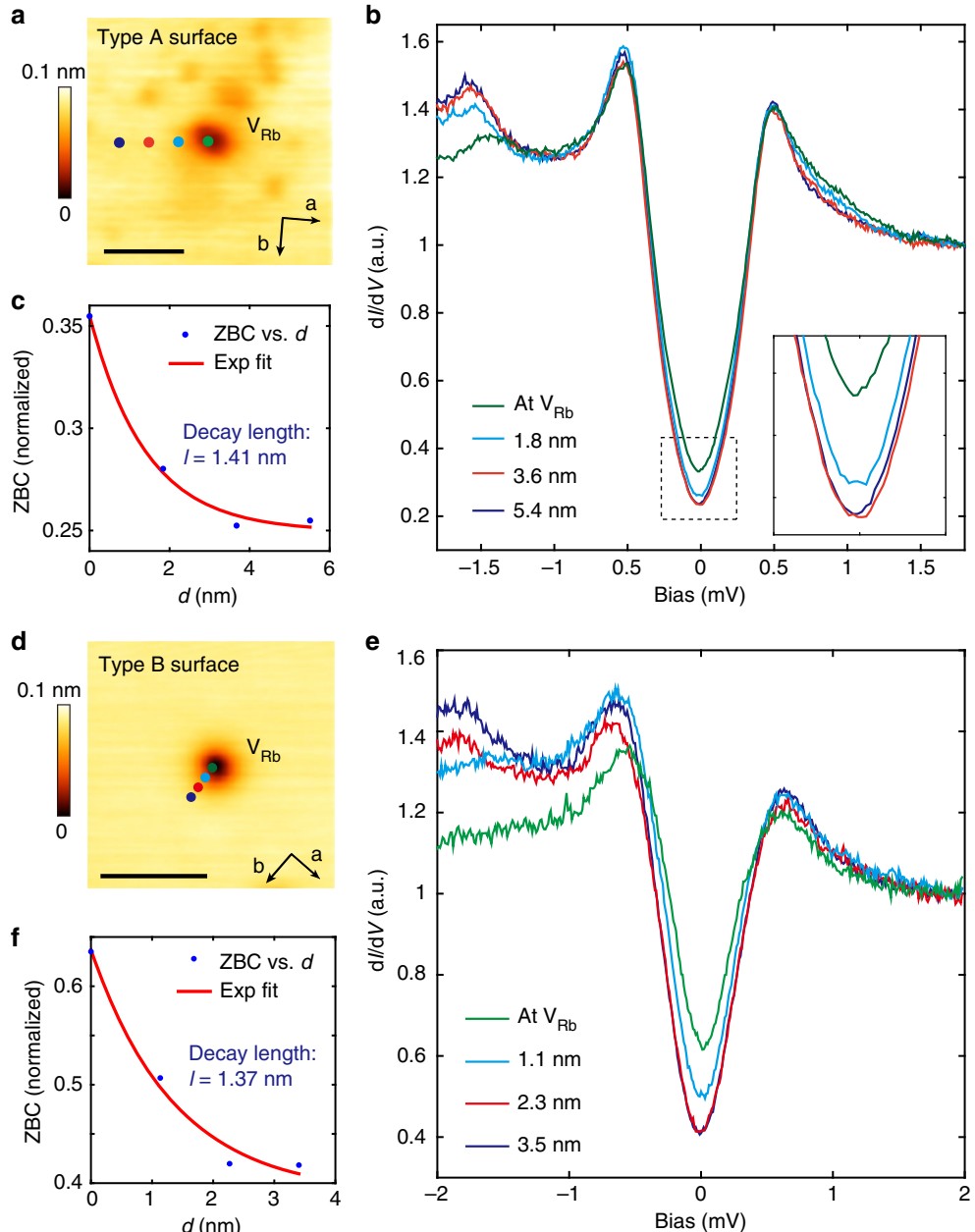

**Fig. 4** Effect of a surface defect ($V_{Rb}$) on superconductivity. **a** Topographic image of a surface Rb vacancy ($V_{Rb}$) on type A surface ($V_b = 30$ mV, $I = 10$ pA; scale bar: 5 nm). **b** $dI/dV$ spectra taken at various distances from $V_{Rb}$ ($V_b = 1.8$ mV, $I = 100$ pA, $\Delta V = 50$ μV, $T_{eff} = 310$ mK), the tip positions are marked in panel **a**. Inset displays the gap bottom showing the suppression of the gap. **c** ZBC value of the gap bottom as function of distance from $V_{Rb}$; the red curve is an exponential fit, which yields a decay length of 1.41 nm. **d** Topographic image of a Rb vacancy on type B surface ($V_b = 1.0$ V, $I = 10$ pA; scale bar: 5 nm). **e** $dI/dV$ spectra taken at various distances from $V_{Rb}$ ($V_b = 2$ mV, $I = 100$ pA, $\Delta V = 50$ μV, $T_{eff} = 310$ mK), the tip positions are marked in panel **d**. **f** ZBC value as function of distance from $V_{Rb}$; red curve is an exponential fit which yields a decay length of 1.37 nm

with a sign-change, Andreev bound states at zero energy will be formed at boundaries perpendicular to the nodal direction, due to the phase change in the quasi-particles' reflection, and decay into the bulk on the scale of coherence length[57]. However, no bound states will form on boundaries perpendicular to the anti-nodal direction (in the ideal case). Experimentally, atomic step edges on the surface can be treated as (weak) 1D boundaries since they carry line scattering potential, and evidence of Andreev bound state has indeed been observed near [110]-oriented step edges in $Bi_2Sr_2CaCu_2O_{8-\delta}$[58].

On type A surfaces we have found [110]- and [100]-oriented step edges formed during cleavage. Topographic images around

these steps are shown in Fig. 5a, c. The orientations of the edges are confirmed by imaging the atomic lattice nearby (e.g. see Fig. 5b), and they are all verified to be single steps with height equal to half of the c-axis lattice constant of $RbFe_2As_2$ (Fig. 5d). Figure 5e, g shows tunneling spectra taken along lines perpendicular to the [100] and [110] edges, respectively. To clearly see the variation of the spectra with distance to the edge, we subtract from them a spectrum taken far away from the step edge and show the difference in Fig. 5f, h. The superconducting gap is suppressed in the immediate vicinity ($d = 0$) of both edges, as indicated by the in-gap peaks in the difference spectra. However, on moving away from the step edge, the decay of the in-gap peaks along [110] is

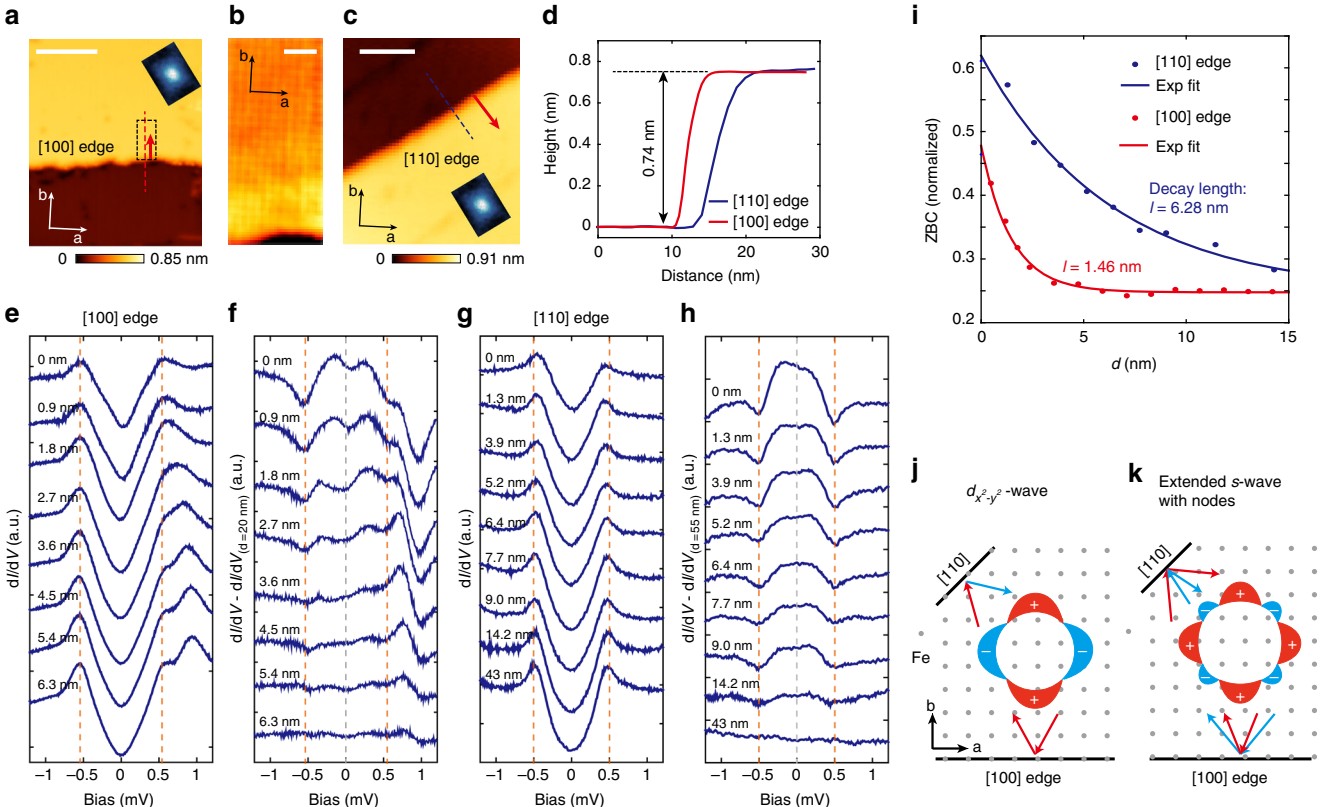

**Fig. 5** Effect of atomic step edges on superconductivity. **a** Topography of a step edge along [100] direction ($V_b = 0.5$ V, $I = 10$ pA, scale bar: 20 nm). **b** Atomically resolved image near the [100] edge ($V_b = 2$ mV, $I = 200$ pA, scale bar: 2 nm), taken in the dashed rectangle in panel **a**. **c** Topography of a step edge along [110] direction ($V_b = 0.5$ V, $I = 10$ pA, scale bar: 20 nm). The orientation of the steps in panel **a** and **c** relative to the $C_4$ symmetry breaking is indicated by the inserted vortex core images. **d** Line profiles along the dashed lines marked in panels **a** and **c**, showing that both steps are half a unit cell high. **e, f** Superconducting gap evolution when leaving the [100] step edge (along red arrow in panel **a**, setpoint: $V_b = 1.5$ mV, $I = 80$ pA, $\Delta V = 50$ μV). A gap far from the step edge has been subtracted for panel **f**. Dashed lines mark the position of coherence peaks and $E_F$. **g, h** Superconducting gap evolution when leaving the [110] step edge (along red arrow in panel of **c**, setpoint: $V_b = 1.5$ mV, $I = 100$ pA, $\Delta V = 50$ μV) and the subtracted spectra. **i** ZBC decay evolution when leaving different steps (dots) and their exponential fits (solid lines). **j, k** Demonstration of the quasi-particle scattering on step edges with different orientation and pairing symmetry. **j** $d_{x^2-y^2}$-wave pairing. **k** extended $s$-wave pairing with eight accidental nodes. (All the data shown in this figure are taken at $T = 20$ mK ($T_{eff} = 310$ mK))

much slower than along the [100] direction. In Fig. 5i, we plot the ZBC values as function of distance from the step edge. The exponential fit yields a decay length of 6.28 nm for the [110] step edge and 1.46 nm for [100]. Such a large (over three times) difference cannot be solely explained by the anisotropic coherence length as reflected in vortex mapping (Fig. 4), since the anisotropic ratio $\xi^L/\xi^S$ of type A surface is only 1.2 (the orientation of the vortex core relative to the step edge is indicated in Fig. 5a, c). It is more likely due to the presence of gap nodes in the {[110]} directions. As shown in Fig. 5j, for $d_{x^2-y^2}$ like pairing, a [110] boundary can give rise to bound states as it is perpendicular to the nodal direction, and these will decay into the bulk on the scale of $\xi$, while the [100] boundary cannot induce such a state. For the extended $s$-wave pairing suggested in[35,43] (Fig. 5k), neither [110] nor [100] boundaries can induce bound states because both are perpendicular to anti-nodal directions. The relatively long decay length for the [110] step edge evidences the formation of bound states (although it is still shorter than the 15 nm $\xi$ for this direction, as discussed below). Gap suppression is not expected for an ideal [100]-oriented edge under $d_{x^2-y^2}$ pairing. However, we notice that the decay length on [100] edge (1.46 nm) is very close to the decay length of gap suppression near $V_{Rb}$ (1.41 nm, see Fig. 4c), which is measured roughly along [100] direction. Thus the gap suppression on [100] edge could be induced by random

disorders near the edge (local disorders always exist on step edges, particularly for the energy-disfavored [100] edge here). We note that the point-like defect induced in-gap states are usually localized within several lattice constant to the defect site[59,60], as theoretically their intensity varies as $1/d^2 e^{-d/\xi}$, gives a decay scale much shorter than $\xi$[59,61].

The above observations are consistent with the $d_{x^2-y^2}$-like pairing suggested by several theoretical works on heavily hole-doped FeSC[17,23]. We also note some recent theoretical work has suggested that electronic nematicity is compatible with a mixing of $s + d$ (or $s + id$) pairing[62,63], which may also be applicable for RbFe$_2$As$_2$. In the mixed state, signatures of $d$-wave pairing should still exist, such as the nodal gap in d$I$/d$V$ and different response at the [110] and [100] edges. However, these signatures could be blurred by the $s$-wave component, e.g., it weakens the intensity of bound states at [110] edges and gives shorter decay length. How these theories would be modified by the $(\pi, \pi)$ diagonal nematicity observed here requires further study.

**Effect of surface potassium (K) dosing on RbFe$_2$As$_2$.** To gain more insights on how the $(\pi, \pi)$ nematic state and superconductivity interplay, we hereby explore how they evolve with doping. We thus performed in situ surface K dosing on RbFe$_2$As$_2$ in an STM system working at $T = 4.5$ K (see Methods for details).

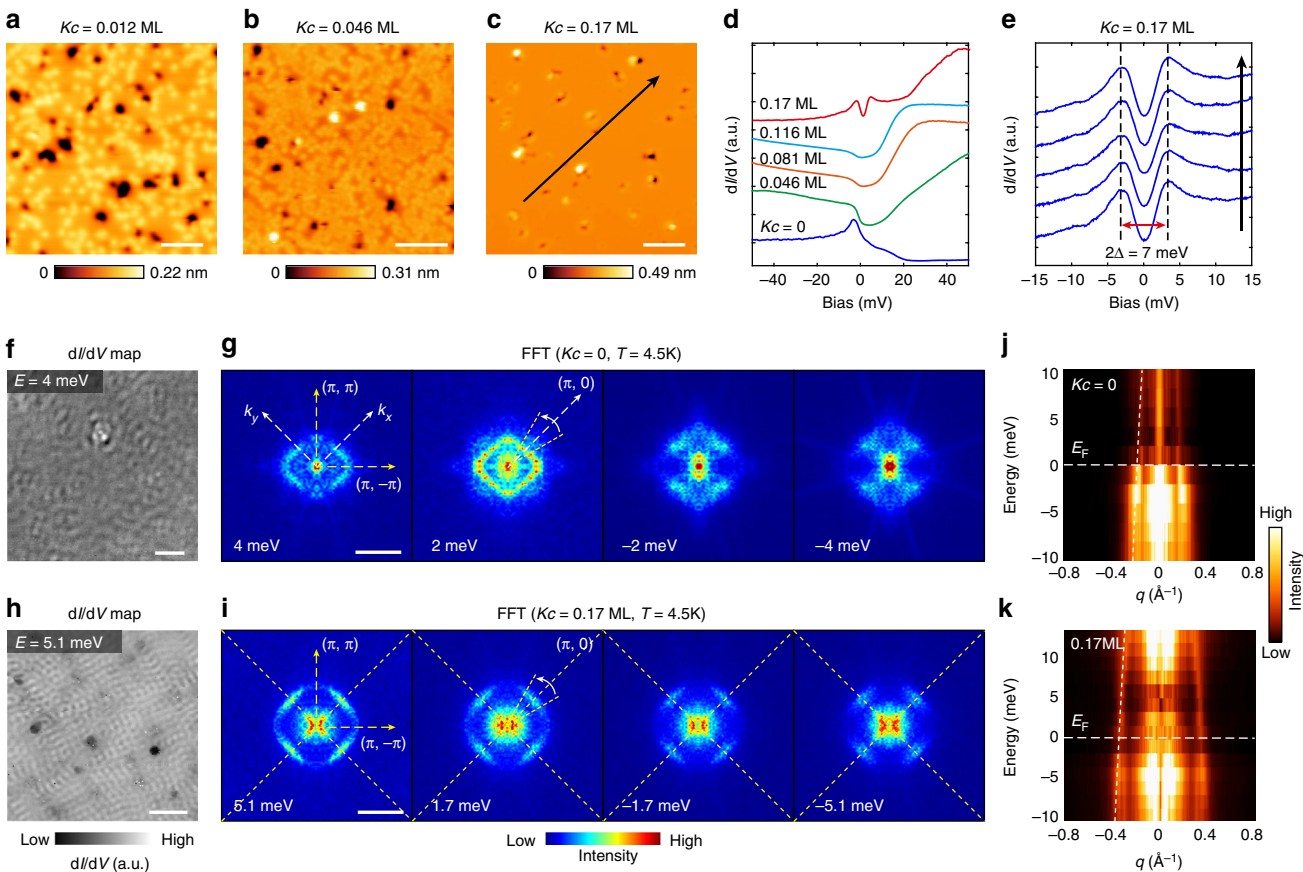

**Fig. 6** Effect of surface potassium (K) dosing on RbFe$_2$As$_2$. **a–c** Topographic images of RbFe$_2$As$_2$ with various K coverage ($Kc = 0.012$ ML, 0.046 ML, 0.17 ML, respectively). K atoms appear as random bright spots at low $Kc$. (Setpoints of **a**: $V_b = 1$ V, $I = 50$ pA; **b**: $V_b = 1$ V, $I = 20$ pA, **c**: $V_b = 0.2$ V, $I = 200$ pA; scale bars: 10 nm). **d** Spatially averaged d$I$/d$V$ spectra of RbFe$_2$As$_2$ with various $Kc$ ($V_b = 50$ mV, $I = 200$ pA, $\Delta V = 1$ mV, $T = 4.5$ K for all the spectra). **e** Low-energy d$I$/d$V$ spectra of RbFe$_2$As$_2$ at $Kc = 0.17$ ML ($V_b = 15$ mV, $I = 300$ pA, $\Delta V = 1$ mV, $T = 4.5$ K), taken along the arrow in panel **c**. A spatially uniform gap with the size of $2\Delta = 7.0$ meV is observed. **f** A representative d$I$/d$V$ map taken at $Kc = 0$, $T = 4.5$ K ($V_b = 10$ mV, $I = 200$ pA, $\Delta V = 1$ mV; scale bar: 10 nm). **g** Selected FFT images of d$I$/d$V$ maps with $Kc = 0$ (scale bar: 0.3 Å$^{-1}$). The FFTs are mirror symmetrized along ($\pi$, $\pi$) and ($\pi$, $-\pi$) directions. **h** A representative d$I$/d$V$ map taken at $Kc = 0.17$ ML ($V_b = 10$ mV, $I = 200$ pA, $\Delta V = 1$ mV, $T = 4.5$ K; scale bar: 10 nm). **i** Selected FFT images of the d$I$/d$V$ maps with $Kc = 0.17$ ML (scale bar: 0.3 Å$^{-1}$); FFTs are also mirror symmetrized along ($\pi$, $\pi$) and ($\pi$, $-\pi$) directions. **j** FFT profiles along the ($\pi$, 0) direction for $Kc = 0$ (averaged over a 30° angle, as indicated in the $E = 2$ meV image in panel **g**). Parabolic fit (dashed curve) gives $E_b = 25$ meV and $q_F = 0.19$ Å$^{-1}$. **k** FFT profiles along the ($\pi$, 0) direction (averaged over a 30° angle) for $Kc = 0.17$ ML. Parabolic fit (dashed curve) gives $E_b = 50$ meV and $q_F = 0.34$ Å$^{-1}$

Dosing K atoms will lower the hole-doping of the (top) FeAs layer via introducing electrons. Figure 6a–c shows typical topographic images with various K coverages ($Kc = 0.012$–0.17 ML, and one monolayer (ML) is defined as the areal density of Fe atoms in a FeAs layer, which is ~13.4 nm$^{-2}$). Without K dosing ($Kc = 0$), anisotropic QPI patterns that breaks ($\pi$, $\pi$)/($\pi$, $-\pi$) equivalence were also observed at $T = 4.5$ K, as shown in Fig. 6f, g (see Supplementary Fig. 9 for additional data), indicating that the ($\pi$, $\pi$) nematicity persists above $T_c$. Upon K dosing, as shown in Fig. 6d, the broad DOS peak below $E_F$ is gradually suppressed as $Kc$ increases, while a gap of $\Delta \sim 3.5$ meV is opened at $E_F$ at $Kc = 0.17$ ML, which is spatially uniform (Fig. 6e). The QPI patterns at $Kc = 0.17$ ML are shown in Fig. 6h, i, which remarkably became rather four-fold symmetric (see Supplementary Fig. 9 for raw FFTs). This indicates that the nematicity is greatly suppressed in the less-hole-doped regime, compatible with the current understanding of the phase diagram. We further checked the temperature dependence of the tunneling gap at $Kc = 0.17$ ML and found it closes at $T \sim 12$ K (Supplementary Fig. 10). Note that for $Kc = 0.17$ ML, the doping of the top FeAs layer is expected to be 0.33 holes/Fe atom

(assuming each K atom can dope one electron, which could be less). Then it will be comparable to Ba$_{1-x}$K$_x$Fe$_2$As$_2$ with x ≥ 0.66 in the hole over-doped region. Thus the observed 3.5 meV gap is most likely a superconducting gap with a $T_c$ of ~12 K (which gives $2\Delta/k_B T_c = 6.8$). Therefore, the above results directly evidence that the ($\pi$, $\pi$) nematicity is suppressed at reduced hole-doping, while the superconductivity is simultaneously enhanced.

The scattering weight in Fig. 6i ($Kc = 0.17$ ML) are particularly strong near the ($\pi$, 0)/(0, $\pi$) directions (dashed lines), suggesting they are from intra-band scattering of a square-shaped pocket. In Fig. 6j, k we plot FFT profiles taken around ($\pi$, 0) directions for $Kc = 0$ and 0.17 ML, respectively, with parabolic fitting applied. The overall dispersion for $Kc = 0$ has a $q_F = 0.19$ Å$^{-1}$ and $E_b = 25$ meV, which is similar to Fig. 2j that measured at $T = 20$ mK ($T_{eff} = 310$ mK) on type A surface. For $Kc = 0.17$ ML, the dispersion is also hole-like but has a significantly larger $q_F = 0.34$ Å$^{-1}$ and $E_b = 50$ meV (Assuming $q_F = 2k_F$ for intra-band scattering, the resulting $k_F = 0.17$ Å$^{-1}$ is comparable with that of ARPES observed $\alpha$ pocket of Ba$_{0.3}$K$_{0.7}$Fe$_2$As$_2$ ($k_F \sim 0.21$ Å$^{-1}$, ref. [37])). However, a reduced $q_F$ at $Kc = 0$ is

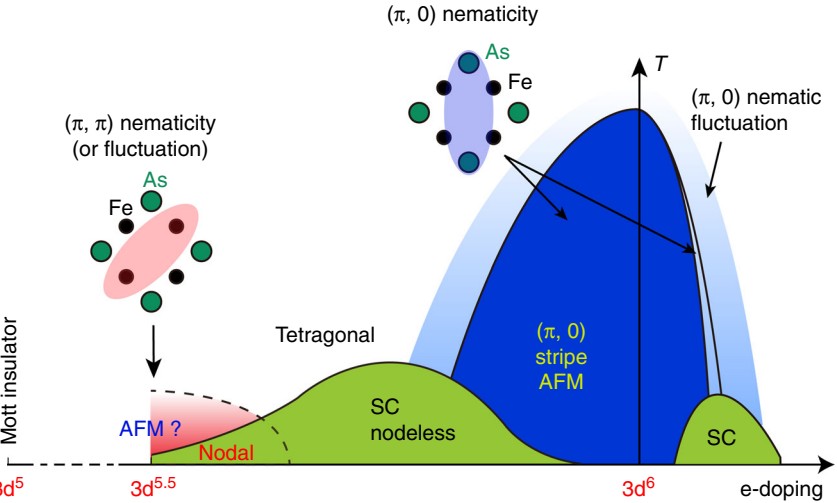

**Fig. 7** Schematic phase diagram of 122 type iron pnictides (based on refs. [1–3,15] and the data of this work). The blue area denotes $(\pi, 0)$ stripe AFM which coexists with the $(\pi, 0)$ nematicity, and the light blue area denotes $(\pi, 0)$ nematic fluctuation. The two green areas are superconducting dome. The red area denotes $(\pi, \pi)$ nematicity at the $3d^{5.5}$ configuration (which may become $(\pi, \pi)$ nematic fluctuation at increased temperature). For the hole-doping superconducting dome, electron pairing is nodeless ($s_\pm$) in the middle of the dome, but became nodal ($d_{x^2-y^2}$ like) as approaching $3d^{5.5}$. A Mott insulator phase is theoretically expected at the $3d^5$ configuration. The upper insets demonstrate the symmetry breaking in $(\pi, 0)$ and $(\pi, \pi)$ nematic state

apparently unexpected, as it has a higher hole concentration than the K-dosed case. It would then imply certain band reconstructions happened in the presence of $(\pi, \pi)$ nematicity at $Kc = 0$.

## Discussions

Our measurement evidences a novel $(\pi, \pi)$ diagonal nematicity that coexists with a nodal $d_{x^2-y^2}$ like pairing component in the strongly hole-doped $RbFe_2As_2$ (with a $3d^{5.5}$ configuration). The diagonal nematicity persists into the normal state above $T_c$, which implies that the superconductivity emerges in the nematic phase, and nematic fluctuations may play a role. Furthermore, the surface electron doping clearly demonstrated an anti-correlation between such a nematicity and superconductivity. In Fig. 7 we summarize above results in a sketch of global phase diagram of 122 type iron pnictides. These results give important clues on the origin of nematicity, and highlight the intimate relation between nematicity and superconducting pairing. As shown in Fig. 7, the stripe AFM and $(\pi, 0)$ nematicity of the parent material weakened through hole-doping, and disappear near the "optimal" doping with highest $T_c$ (refs. [1,2,15]). The pairing at optimal doping is widely believed to be $s_\pm$ wave and mediated by $(\pi, 0)$ spin fluctuations[3,4]. Here in the over-hole-doped regime, the re-emergence of a new type of diagonal nematic state and change in the pairing symmetry with suppressed $T_c$ imply the system approaches a new regime with different fluctuations. Since an AFM Mott insulting state resembling the parent compound of cuprates was predicted for $3d^5$ configuration[18,19], the related spin/nematic fluctuations are likely the candidate. It may give rise to or mediate both the diagonal nematicity and the $d$-wave pairing component, although the microscopic details require further theoretical refinement. Therefore, our results would lay the groundwork for a unified understanding of the cuprates and FeSCs.

While preparing this paper, we became aware of an NMR study on $CsFe_2As_2$, which also suggests a nematic state along the $(\pi, \pi)$ direction[64]. It thus provided complementary evidences for the re-emergence of electronic nematicity in heavily hole-doped FeSCs. We also noticed that another NMR study[65] has revealed strong spin fluctuations in $RbFe_2As_2$ and $CsFe_2As_2$.

## Methods

**Sample growth and transport measurements**. The $RbFe_2As_2$ singles crystals were grown in alumina crucibles by a self-flux method, as described in ref. [28]. Zero resistance is observed below $T_c \sim 2.5$ K (Supplementary Fig. 1c). By fitting the resistance data with $R = R_0 + kT^2$ for $T \leq 50$ K (Supplementary Fig. 1b), the residual resistivity ratio (RRR $= R_{(T=300\,K)}/R_0$) is calculated to be 533. Such a large RRR value has been reported for $AFe_2As_2$ ($A = $ K, Rb, Cs), which was attributed to a strong correlation induced coherence–incoherence crossover at high temperatures[27].

**STM measurements and surface K dosing**. Low-temperature STM experiment (except the surface K dosing effect) was conducted in a commercial $^3$He/$^4$He dilution refrigerator STM (Unisoku) at the base temperature of ~20 mK. The effective electron temperature ($T_{eff}$) of this system was checked to be ≤310 mK by measuring the superconducting gap of Al films (see Supplementary Note 1). $RbFe_2As_2$ samples were cleaved in ultrahigh vacuum at ~80 K (liquid nitrogen temperature) and immediately transferred to the STM module. Pt tips were used after careful treatment on Au (111) sample. The tunneling spectroscopy ($dI/dV$) was performed using a standard lock-in technique with modulation frequency $f = 787$ Hz, and the modulation amplitudes ($\Delta V$) are specified in the figure captions.

The surface K dosing on $RbFe_2As_2$ was conducted in another cryogenic STM at $T = 4.5$ K. $RbFe_2As_2$ sample was cleaved at ~30 K and Pt tips were also used after treatment on Au (111). K atoms were evaporated from standard SAES alkali metal dispensers, and the sample was kept at 80 K during deposition. The deposition rate was carefully calibrated by directly counting surface K atoms at low coverage. The tunneling spectra were obtained by using lock-in technique with modulation frequency $f = 915$ Hz and amplitude $\Delta V = 1$ mV.

## Data availability

All the original data related to this study are available from the corresponding author upon reasonable request.

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

## Acknowledgements

We thank Professor P. Adelmann for the help on sample growth, and thank Professors T. Wu, J. P. Hu, D. H. Lee and Dr. D. Peets for helpful discussions. This work is

supported by the National Natural Science Foundation of China, National Key R&D Program of the MOST of China (Grant No. 2016YFA0300200 and 2017YFA0303004), National Basic Research Program of China (973 Program) under grant No. 2015CB921700, and Science Challenge Project (grant no. TZ2016004).

## Author contributions

The growth of RbFe$_2$As$_2$ singles crystals was performed by T.W. The STM measurements and data analysis were performed by X.L., R.T., M.R., W.C., Q.Y., Y.Y. and T.Z.. T.Z. and D.L. Feng coordinated the project and wrote the manuscript. All authors have discussed the results and the interpretation.

## Additional information

**Competing interests:** The authors declare no competing interests.

