## [Peer Review File · Nature Communications]

Reviewers' comments:

Reviewer #1 (Remarks to the Author):

This manuscript uses scanning tunneling microscopy to investigate the hole-doped superconductor RbFe₂As₂ with T_c=2.5K. The authors make 2 main claims:

- (1) They use quasiparticle interference imaging, and vortex core state imaging, to show an unusual nematicity (rotational symmetry breaking) along the diagonal direction, 45 degrees away from the Fe-Fe bond direction. This contrasts with most other Fe-based superconductors that show nematicity along the Fe-Fe bond directions.
- (2) They image the different decay length of in-gap states near step edges along the (110) vs (100) surface directions to argue that the superconducting pairing order parameter is dx²-y².

Regarding claim (1): The data clearly justifies this claim. As far as I know, the 45deg nematicity is unique, but the authors cite a large group of references (24-41) all lumped together, and I didn't have time to read them all, and it's hard to tell from their wording whether they are claiming that their experiment is the first time that 45deg nematicity has been seen in any Fe-based superconductor.

Regarding claim (2): The evidence in this manuscript is not strong. However, such dx²-y² pairing has already been seen in closely related compound KFe₂As₂, so the claim is believable, albeit not remarkable.

I marked up the PDF with a number of small suggestions to improve the clarity of the paper. I copy the most important points here:

(1) I would like more clarity about the difference between surfaces A & B.

- a. How many different FOVs have you imaged? On how many distinct samples? If you histogram the defect densities in the different FOVs that you have imaged, are they really bimodal, or is there a continuum between surface A & surface B?
- b. You mention that you think surface B is strained. Can you apply a sensitive "spatial lockin" technique (e.g. Lawler, Nature 2010, 10.1038/nature09169) to quantify your speculated strain? Note that you would need access to both surfaces A & B with the same tip, in order to calibrate this properly.

(2) I would like more clarity about the temperature calibration.

- a. What is the superconducting coherence length of Al? (e.g. reference: 10.1103/PhysRevB.5.3558 shows that it varies with film thickness) It looks like your island is only ~30 nm big, which is much smaller than the bulk coherence length, $\xi > 1\mu\text{m}$. Furthermore, the T_c (& therefore the gap) of Al films increases with decreasing film thickness. So it doesn't seem like nano-islands of thin films of Al are a very good calibration for your electron temperature. Why not use a bulk sample with well-known T_c?
- b. In supplemental figure S2, how are you disentangling the effects of T_{eff} and Gamma? I wouldn't expect that Gamma is actually 0, because you must have some non-thermal broadening from the fact that your island is so small.

(3) On page 4, you claim that QPI is isotropic above 7 meV, but anisotropic below 4 meV. What is the significance of this energy scale around 4-7 meV? According to Fig 1e and Figs 2e,i,j:

- 0.46 meV = SC gap
- 24 meV = top of one hole band on surface A
- 30 meV and -4 meV = tops of 2 different hole bands on surface B

So I don't see why anything special should happen in the 4-7 meV range.

(4) Regarding the spectra in Fig 1f: Please clarify if these are "typical" or spatially averaged in defect-free regions, or spatially averaged over whole FOV... it would be helpful to have a supplemental figure that gives a better idea of the spatial variation of these spectra on each of the two different surfaces.

(5) Regarding Fig 2:

- a. It looks like you have done some symmetrization in panels c & h. Please be clear about exactly what analysis you have done on the raw data.
- b. Panel e: The FFT is so clearly anisotropic in panel (c), so I don't understand what this azimuthal average means. Azimuthal averaging makes sense to me when you have an isotropic signal and you want to increase signal-to-noise. But when you have an anisotropic signal, it doesn't make sense to azimuthally average.
- c. Panels i,j: If it's along a specific direction such as (π,π) then it's not azimuthally averaged, and I don't understand what it's averaged over.

(6) Fig 4: please state the setup conditions for the spectra. The signal that you're pointing out – the variation in zero bias conductance – is very weak, so it's very important to rule out setup condition artifacts. It would be better to show a full map of the zero bias conductance over this defect, with clearly stated setup condition.

(7) Fig 5h: I don't understand the colors of the arrows.

- a. Left panel: Shouldn't left-going arrow be blue, and down-going arrow be red, to match the directions of the gaps drawn on the circle? Meanwhile the diagonal arrows are along nodes, so their sign (color) is undefined?
- b. Right panel: I don't understand why the left and down arrows are blue: shouldn't they be red? And the diagonal arrows should be blue?

I am not yet convinced that this paper is of sufficient general interest for the wide readership of Nature Communications. It discusses a very specific finding in a very low- T_c superconductor. The last sentence is rather generic and doesn't actually clarify at all how they are laying any groundwork for a unified understanding: "Our results give important clues on the origin of nematicity and lay the groundwork for a unified understanding of the cuprates and FeSCs."

Reviewer #2 (Remarks to the Author):

In this MS the authors show their STM results, which, they argue, indicates the existence of the novel nematic order in strongly hole-doped RbFe_2As_2 . Interestingly, they found that the nematic order breaks the symmetry between (p,p) and $(p,-p)$ directions rather than between $(p,0)$ and $(0,p)$ direction, as in other Fe-based materials. They also argue that the superconducting gap is nodal and possibly has dx^2-y^2 symmetry.

The results are interesting and worth publishing. However, I am not positive that they meet Nat. Comm. criteria by the following reasons:

1. This is not the first report of possible diagonal nematic order in strongly hole-doped Fe-pnictides. Earlier evidence was presented in Ref. [63] back in 2016.
2. The diagonal nematic order should give rise to elongation of the hole pocket along either (p,p) or $(p,-p)$ direction. The authors supposedly extracted the dispersion from STM, but didn't show the actual

shape of the hole pocket. For comparison, STM study FeSe demonstrated that a hole pocket is elongated along Y direction [Sprau et al, Science 357, 6346 (2017)], consistent with axial nematic order.

3. The authors wrote in the abstract "Our results highlight the intimate connection between nematicity and superconducting pairing". I didn't find this connection. The authors discuss possible d-wave gap symmetry, but its relation to diagonal nematicity (if any) is unclear.

A few additional remarks:

1. On p. 2, the authors related d-wave gap symmetry with (p,p) spin fluctuations. How this is related to RbFe₂As₂ in which neutron scattering and NMR data (see, e.g. Ref. [64]) show that spin fluctuations are nearly stripe-type (i.e., the peak is at $(p(1+\delta),0)$ and related points?

2. What is "stripe-type AFM (p,p) order" (p. 5 of the MS), and how it can promote diagonal nematicity? More generally, the authors wrote in the discussion that (p,p) spin fluctuations are likely candidate for promoting a diagonal nematicity. How this is possible if, as the authors stated, only one hole pocket at Γ has been observed in 2Fe zone?

Reviewer #3 (Remarks to the Author):

The manuscript of Liu et al present a study of superconductivity and the electronic structure in RbFe₂As₂ by low temperature scanning tunnelling microscopy and spectroscopy and quasi-particle interference imaging. The results are novel and interesting, and in principle suitable for publication in Nature Communications. There are a few issues the authors should address however prior to publication.

The authors claim that they have discovered a novel type of nematicity which is rotated by 45deg compared to that of other iron pnictides, though similar to the electronic anisotropy seen in FeTe. The interpretation hinges on the assignment of the atomic structure of the surface: the authors assign the atomic resolution to a $\sqrt{2}\times\sqrt{2}$ reconstruction of the surface, in which case the symmetry breaking differs indeed by 45deg from that of other pnictides. If, however, one would assign the atomic resolution to either a complete Rb layer or the As layer, the electronic anisotropy would be the same as in other pnictides. Because the interpretation hinges on the assignment of the atomic resolution, this point deserves further clarification by the authors:

- is there any other evidence (apart from the images shown in the manuscript) that the surface exhibits a $\sqrt{2}\times\sqrt{2}$ reconstruction? (e.g. LEED? Cleaves which result in different terminations?)
- If the Rb reconstructs into a $\sqrt{2}\times\sqrt{2}$ order, this must mean that diffusion takes place between cleavage and imaging. Have the authors tried to cleave at lower temperatures to prevent diffusion from occurring?
- there should be domains of $\sqrt{2}\times\sqrt{2}$ order. Have the authors seen evidence of these domains?

The atomic distances are not a very good indicator for the surface termination, as the calibration will change with temperature (which can be particularly an issue in a millikelvin system) – unless the authors have calibrated their instrument independently at the same temperature as the topographic images shown, this can yield misleading assignments.

Minor issues:

- There is an issue with the Dynes function, eq. 1 – the modulus should be outside of "Re".

- What is the setpoint condition used for the maps in fig. 2? Is it the same in both cases? What lock-in modulation was used for the maps?
- Is it clear that figs. 1a and b are different terminations? What is the evidence for this? If it is spectroscopic, could it just be different tips sensitive to different orbitals? (as seen, e.g., in FeSe, see Science 357, 75). From the crystal structure, there doesn't seem to be an obvious way to have two different surface terminations which both have the same $\sqrt{2} \times \sqrt{2}$ superstructure of Rb.
- Why does the resistance drop below zero? (fig. S1c)
- Pg. 4: re the statement that "Such C4 symmetry breaking between (n, n) and $(n, -n)$ has never been reported for other iron pnictides." This is true for pnictides, but not for chalcogenides (i.e. FeTe). This is briefly discussed in the discussion section, but could be mentioned earlier in the introduction.
- In the discussion of the relation between nematicity and superconductivity, the authors may also want to compare their work to Sci. Adv. 1, e1500206, which has investigated this relation for FeSeTe.
- Fig. 1: please provide temperature at which this has been taken
- Caption of fig. 2, please indicate whether images are symmetrized (this is clear from the supplementary, but should be also clearly stated in the caption in the main manuscript)
- Please provide setpoint conditions for all topographic images and setpoint condition and lock in parameters for all spectroscopic maps and spectra (also in supplementary material). Also temperatures of the STM should be indicated for all measurements (not just a noise temperature), unless all have been taken at 20mK (in which case this should be indicated as well).

Reply to reviewers:

We thank all the reviewers for their time and insightful comments on our manuscript. Our responses are in blue text below and the original comments are in italic. The corresponding revisions in the manuscript are highlighted.

Reviewer 1:

We thank Reviewer 1 for the elaborative review of our manuscript and thoughtful comments which greatly helped us to improve the clarity and strengthen the conclusion. We have addressed all the comments including that given in the annotated PDF file, as listed below.

This manuscript uses scanning tunneling microscopy to investigate the hole-doped superconductor $RbFe_2As_2$ with $T_c=2.5K$. The authors make 2 main claims:

- (1) They use quasiparticle interference imaging, and vortex core state imaging, to show an unusual nematicity (rotational symmetry breaking) along the diagonal direction, 45 degrees away from the Fe-Fe bond direction. This contrasts with most other Fe-based superconductors that show nematicity along the Fe-Fe bond directions.*
- (2) They image the different decay length of in-gap states near step edges along the (110) vs (100) surface directions to argue that the superconducting pairing order parameter is $d_{x^2-y^2}$.*

Regarding claim (1): The data clearly justifies this claim. As far as I know, the 45deg nematicity is unique, but the authors cite a large group of references (24-41) all lumped together, and I didn't have time to read them all, and it's hard to tell from their wording whether they are claiming that their experiment is the first time that 45deg nematicity has been seen in any Fe-based superconductor.

We thank reviewer 1 for pointing out the potential importance of the paper. Before this work, 45° or diagonal nematicity had not been observed in all the iron-pnictides and most iron-chalcogenides. FeTe is a special exception in which a 45° symmetry breaking was observed in QPI (ref. 52). It is likely due to pure FeTe has a unique bicollinear AFM order with $Q = (\pi/2, \pi/2)$. However upon doping the magnetism of FeTe changes drastically and the “normal” $(\pi, 0)$ nematicity recovered (such as that observed in FeSe and $FeTe_{1-x}Se_x$). Our work report the first observation of diagonal nematicity in iron-pnictides, which coexists with superconductivity and displays an anti-correlation with it. Related discussion can be found at page 6, paragraph 2 and page 9, paragraph 3 of the main text. As we noted at the end of the manuscript, we became aware of a NMR preprint (ref. 63) on diagonal nematicity in $CsFe_2As_2$ when writing up this work, but NMR study are less direct compared with STM.

Refs. 24-41 are previous experimental studies on AFe_2As_2 ($A=K, Rb, Cs$). We have cited them separately throughout the paper when necessary. Refs. 5-15, 47-53, 61-63 are relate to the nematicity of Fe-based superconductors.

Regarding claim (2): The evidence in this manuscript is not strong. However, such $d_{x^2-y^2}$

y₂ pairing has already been seen in closely related compound KFe₂As₂, so the claim is believable, albeit not remarkable.

We agree with reviewer's judgment on the data.

I marked up the PDF with a number of small suggestions to improve the clarity of the paper. I copy the most important points here:

(1) *I would like more clarity about the difference between surfaces A & B.*

a. *How many different FOVs have you imaged? On how many distinct samples? If you histogram the defect densities in the different FOVs that you have imaged, are they really bimodal, or is there a continuum between surface A & surface B?*

We had imaged about 15 different surface regions on four distinct samples (came from the same batch). Most regions have a defect (V_{Rb}) density of $0.02 \sim 0.04 \text{ nm}^{-2}$, showing moderate $(\pi, \pi)/(\pi, -\pi)$ symmetry breaking similar to Fig. 2c and we refer them as surface A. Enhanced symmetry breaking that similar to Fig. 2h was observed on two different regions in one sample, which have a defect density of $0.01 \sim 0.02 \text{ nm}^{-2}$ and we refer them as surface B. We cannot tell if the defect densities are really bimodal due to limited samples (it seems unlikely), and we did not happen to observe a transition area between type A and B surfaces. However, we do not think V_{Rb} defects play an important role in making type A and B surfaces different. Firstly, their absolute densities are small. V_{Rb} could dope holes to the underlying FeAs layer. If assuming each V_{Rb} dopes one hole, then for type A surface each Fe atom will acquire $0.0015 \sim 0.003$ holes, and for type B surface the value is $0.0008 \sim 0.0015$ holes/Fe. Such additional dopings are less than 1% of the original doping of RbFe_2As_2 (0.5 hole/Fe), and thus it is unlikely to make any significant difference. Secondly, the V_{Rb} defect only show weak suppression to the superconducting gap, within a short length scale of $\sim 1.4 \text{ nm}$ (see Fig. 4 and Fig. S9 of the revised manuscript). The overall broadening of the gap on type B surface is unlikely due to impurity effect, and we speculate it could be related to enhanced nematicity in type B surfaces.

Therefore, we think type A and B surfaces are not different surfaces with distinct defect densities. They have the same lattice structure and are just differed by the strength of the nematicity which may be sensitive to strain. We feel that refer them as "surface A" and "surface B" may be a kind of misleading. In the revised manuscript we renamed them as "type A surface" and "type B surface".

b. *You mention that you think surface B is strained. Can you apply a sensitive "spatial lockin" technique (e.g. Lawler, Nature 2010, 10.1038/nature09169) to quantify your speculated strain? Note that you would need access to both surfaces A & B with the same tip, in order to calibrate this properly.*

We thank reviewer for pointing out a possible way to characterize the strain and nematicity. Lawler *et al.* used this method in Nature 466, 347(2010) to quantify the intra-unit-cell nematicity in cuprates (which appears as non-dispersive modulation that locally breaks the C_4 - C_2 symmetry). Such a method measures the difference

between unequivalent Bragg peaks in the FFT image, and it is sensitive to anisotropic electronic structures induced by nematicity or strain on a symmetric lattice. We noticed that this technique needs high precision, atomically resolved dI/dV mapping, and it mainly detects the non-dispersive states. Unfortunately, so far we had not achieved such atomically resolved dI/dV mapping on type A and B surfaces. And as the reviewer mentioned, one will need exactly the same tip states to access different sample surfaces, which makes it more challenging since the chance of finding type B surfaces is quite low. Therefore, at current stage we are not able to apply this technique. Nevertheless, we think quantifying the strain or nematicity using similar technique certainly deserves more study in the future.

(2) I would like more clarity about the temperature calibration.

a. What is the superconducting coherence length of Al? (e.g. reference: 10.1103/PhysRevB.5.3558 shows that it varies with film thickness) It looks like your island is only ~30 nm big, which is much smaller than the bulk coherence length, $\xi > \lambda_{um}$. Furthermore, the T_c (& therefore the gap) of Al films increases with decreasing film thickness. So it doesn't seem like nano-islands of thin films of Al are a very good calibration for your electron temperature. Why not use a bulk sample with well-known T_c ?

The superconducting coherence length of Al (bulk) is $\sim 1.55\mu\text{m}$ (C. Poole, *Handbook of Superconductivity*, Academic Press (2000), Page 436). We understand reviewer's concern about the limited size of the island in Fig. S2a. In fact, Fig. S2a is only a very local region of the film ($75 \times 75 \text{ nm}^2$). In Fig. R1 below we show a large scale image of the same film (size: $250 \times 250 \text{ nm}^2$). One can see although the film is not perfectly flat, it is *continuous* with terraces and island structures. The island structures are not isolated grains, so their superconductivity should not be limited by their apparent size. Indeed, the T_c of *granular* Al films had been shown to increase as the thickness decrease (e.g. B. Abeles *et al*, PRL, **17**, 632(1966)), possibly due to "surface superconductivity" (V. L. Ginzburg, Phys. Letters **13**, 101 (1964)). However, it was also reported that for crystalline Al films, which correspond to our case, do not show significant T_c enhancement comparing to bulk Al (M. Strongin *et al.*, *Low Temperature Physics-LT 13*, pp 563-567, Springer, New York 1974). In the revised manuscript we have replaced Fig. S2a with Fig. R1.

We use Al films to check electron temperature just because we have an MBE chamber connected with the STM system, and we can easily grow fresh films with clean surfaces. We noticed a recent paper (H. Baek, *et al.*, PRB 92, 094510 (2015)) also used Al thin film to calibrate T_{eff} of a millikelvin STM (see its page 8, Fig. 9). As the referee suggested, bulk Al crystals or other bulk superconducting metals are certainly good candidates. We appreciate this suggestions very much, and we will certainly make an effort in the future in this direction. As it will take indefinite time for us to gain experience to obtain their clean surfaces; and due to the limited time, we have to stick to the current method of estimating T_{eff} , however, we think the possible error in it will not affect the conclusions in this work.

Figure R1 | Large scale STM image of 20 ML Al film on Si(111). ($V_b = 3V$, $I = 10$ pA)

b. In supplemental figure S2, how are you disentangling the effects of T_{eff} and Γ ? I wouldn't expect that Γ is actually 0, because you must have some non-thermal broadening from the fact that your island is so small.

Yes the gap should always have finite non-thermal broadening. Here because the measured gap is not very sharp (T_{eff} is not low enough), we actually cannot disentangle T_{eff} and other broadening factors (the Γ term in general). Therefore, to make a conservative estimation of the T_{eff} , we just used T_{eff} to account for *all* the broadening effects. Thus the real T_{eff} of our system could be slightly lower than 310 mK. To give an estimation of non-thermal broadening effect, we noticed the paper (PRB 92, 094510 (2015)) fitted the gap of Al film with a modified Maki BCS theory (K. Maki, Prog. Theor. Phys. 31, 945 (1964)), and they got a depairing factor of $\zeta = 0.019$ meV (which plays similar role as Γ in Dynes formula). If we assume our film has a similar ζ of ~ 0.02 meV. A Maki fitting to our gap will yield $T_{eff} = 290$ mK, as shown in Fig.R2 below. Since we cannot exactly determine the ζ and it doesn't make a large difference on T_{eff} , we prefer to conservatively use $T_{eff} = 310$ mK as our electron temperature. We have added related statement in the revised supplementary materials (page 2, para 1, line 7~9).

Figure. R2 | Red curve: Maki BCS fit to the superconducting gap spectrum (blue dots) of Al film on Si(111). Fitting parameters: $\Delta = 0.19$ meV, $\zeta = 0.02$ meV, $T_{eff} = 290$ mK.

(3) *On page 4, you claim that QPI is isotropic above 7 meV, but anisotropic below 4 meV. What is the significance of this energy scale around 4-7 meV? According to Fig 1e and Figs 2e,i,j:*

- 0.46 meV = SC gap
- 24 meV = top of one hole band on surface A
- 30 meV and -4 meV = tops of 2 different hole bands on surface B

So I don't see why anything special should happen in the 4-7 meV range.

Here we just mean the symmetry breaking happened (or became significant) only at low energies (within 4 meV ~ -7 meV), for type A surface. For type B surface it can be seen in a bit wider range (20 meV ~ -10 meV). The reason for why they can only be observed in such range is unclear for now, since we don't exactly know the driven mechanism of the symmetry breaking. The 4-7 meV range is not special or related to any energy scales that reviewer mentioned above. We have revised this part (page 4, paragraph 2, line 6~8) and removed possible misleading statements.

(4) *Regarding the spectra in Fig 1f: Please clarify if these are "typical" or spatially averaged in defect-free regions, or spatially averaged over whole FOV... it would be helpful to have a supplemental figure that gives a better idea of the spatial variation of these spectra on each of the two different surfaces.*

We have added Fig. S6 in the supplementary materials to show the spatial dependence of superconducting gap. The gaps are quite uniform in the defect-free area of both type A and B surfaces, so we picked typical ones for the gap fittings shown in Fig. 1f.

(5) *Regarding Fig 2:*

a. It looks like you have done some symmetrization in panels c & h. Please be clear about exactly what analysis you have done on the raw data.

Yes the FFT in Figs. 2c, 2h are symmetrized. We first identify the (π, π) / $(\pi, -\pi)$ directions of the raw FFT by using atomically resolved images. Then the raw FFT are mirror symmetrized along the (π, π) and $(\pi, -\pi)$ directions. Detailed process is 1): mirror flip the raw FFT along (π, π) and add it to the raw FFT; 2): flip the results of 1) along $(\pi, -\pi)$ and add it to the results of 1). We have specified this symmetrization process in the revised supplementary materials (page 6, section V).

b. Panel e: The FFT is so clearly anisotropic in panel (c), so I don't understand what this azimuthal average means. Azimuthal averaging makes sense to me when you have an isotropic signal and you want to increase signal-to-noise. But when you have an anisotropic signal, it doesn't make sense to azimuthally average.

We understand reviewer's concern. We made azimuthal average here is just want

to show the “overall” change of the FFT as function of energy. One can see that the evolution of the FFT pattern in Fig. 2c (and Fig. S8) is rather complicated. It was elongated along $(\pi, -\pi)$ at $E > 0$ and switched to (π, π) at $E < 0$. Such behavior should arise from distorted band(s) (ref.49-51), but at this stage it’s difficult to interpret them quantitatively due to the lack of detailed knowledge on band distortion. Nevertheless, the overall size of FFT pattern still changes with energy, indicating that they are originated from QPI. We agree that making a whole azimuthal average is still a kind of confusing. In the revised manuscript, we replaced Fig. 1e with the FFT profiles taken along the k_x (i.e., $(\pi, 0)$) direction (averaged over a 30° angle), since the scattering weights of most FFTs are concentrated around this direction. In the new Fig. 1e an overall hole-like dispersion is still observed, and a parabolic fit gives $E_b = 27$ meV and $q_F = 0.21 \text{ \AA}^{-1}$ (similar to previous value).

c. Panels 1j: If it's along a specific direction such as (π, π) then it's not azimuthally averaged, and I don't understand what it's averaged over.

To enhance the signal/noise, the FFT line cuts in Figs. 2i and 2j are averaged over a 30° angle around the (π, π) and $(\pi, -\pi)$ directions, respectively. We have specified this and marked the angle in the revised Fig. 2h.

(6) *Fig 4: please state the setup conditions for the spectra. The signal that you’re pointing out – the variation in zero bias conductance – is very weak, so it’s very important to rule out setup condition artifacts. It would be better to show a full map of the zero bias conductance over this defect, with clearly stated setup condition.*

The spectra in Fig. 4b are all taken at the same setpoint of $V_b = 1.8$ mV, $I = 100$ pA ($\Delta V = 50 \mu\text{V}$) and normalized by the dI/dV value at $V_b = 1.8$ mV. Since the setpoint energy is much larger than the gap size, the ZBC value is expected to reflect the gap variation. We did not carry out a full map on the defect. Instead, we can show the gap variation near a V_{Rb} defect on type B surface (Fig. S9), similar gap suppression was also observed.

(7) *Fig 5h: I don't understand the colors of the arrows.*

a. Left panel: Shouldn't left-going arrow be blue, and down-going arrow be red, to match the directions of the gaps drawn on the circle? Meanwhile the diagonal arrows are along nodes, so their sign (color) is undefined?

b. Right panel: I don't understand why the left and down arrows are blue: shouldn't they be red? And the diagonal arrows should be blue?

In Fig. 5h some color and orientation of the arrows were not properly chosen, we thank reviewer for pointing this out. We have modified them in the revised manuscript. In the left panel of Fig. 5h, the colors of the arrow at $[110]$ edged are corrected, as they stand for the gap sign on their direction. The arrows at $[100]$ edge are adjusted to have a smaller incident angle (so they are not along nodal direction). In the right

panel, we draw two sets of arrows near both [110] and [100] edges. The arrows with small incident angles have the same color with the gap perpendicular to the edge, and the color (gap sign) changes for larger incident angles.

I am not yet convinced that this paper is of sufficient general interest for the wide readership of Nature Communications. It discusses a very specific finding in a very low- T_c superconductor. The last sentence is rather generic and doesn't actually clarify at all how they are laying any groundwork for a unified understanding: "Our results give important clues on the origin of nematicity and lay the groundwork for a unified understanding of the cuprates and FeSCs."

RbFe₂As₂ is a member of AFe₂As₂ (A=K,Rb,Cs) family, which are the most heavily hole-doped FeSCs. They are of fundamental importance for a complete understanding of the phase diagram of FeSC. Their unique properties such as enhanced electron correlation, orbital-selective Mott transition and nodal pairing have been evidenced in previous studies. However, microscopic study on their electronic structure and pairing symmetry is still lacking. Our work is the first reported QPI measurement and the first phase-sensitive STS measurement on an AFe₂As₂ family member. Furthermore, in this revision we performed additional experiment on RbFe₂As₂. We found that the diagonal nematicity persists into the normal state above T_c , which implies the superconductivity forms out of this state and nematic fluctuations may play a role. Via surface K-dosing to RbFe₂As₂ (Fig. 6). We found the diagonal nematicity is suppressed upon electron doping and the superconductivity is significantly enhanced, which suggests an anti-correlation between them. These results give important clues on the origin of diagonal nematicity and its relation with superconducting pairing. It is known that the $(\pi, 0)$ nematicity of lightly doped FeSCs is closely related to $(\pi, 0)$ stripe AFM order; while in optimal doped FeSC the $(\pi, 0)$ nematicity is substantially weakened and the electron pairing is believed to be s_{\pm} wave and mediated by $(\pi, 0)$ spin fluctuations. Here in the over-hole-doped regime, the re-emergence of a new type of nematicity and change in the pairing symmetry with suppressed T_c imply the system approaches a new regime with different fluctuations. As theoretically expected, an AFM Mott insulating state at $3d^5$ configuration is likely the candidate. The related spin/nematic fluctuations may give rise to (or mediate) both the diagonal electronic nematicity and the d-wave pairing component, although the microscopic details certainly require further theoretical study. Therefore, we think the work would lay the groundwork for a unified understanding of the cuprates and FeSCs. We added more related discussions in the revised manuscript (page 9, paragraph 3, highlighted).

Reply to additional comments in the annotated PDF file:

Page 1:

Comment 1: Do you mean at [100] step edges? Need some descriptor, to distinguish from the subsequent sentence about enhancement.

Reply: here we mean the gap is suppressed on both [100] and [110] step edges, but the suppression is particularly strong at the [110] oriented edges (due to the possible Andreev bound state). We have revised this sentence to make it clear.

Comment 2: Do you mean the gap is particularly strong (i.e. the gap is enhanced), or do you mean it is particularly strongly suppressed?

Reply: We mean the gap suppression is particularly strong at the [110] oriented edges. We have clarified the statement.

Comment 3: What do you mean by diagonal? [110]? Please be very clear & specific.

Reply: Yes we mean the [110] directions. We have clarified the statement.

Comment 4: Please establish at the beginning what is your unit cell: 1 Fe or 2 Fe? Ideally, refer to a schematic figure right away. There are so many conflicting conventions in the literature, so it's hard to know a priori whether (0, pi) means the Fe-Fe direction or Fe-As direction

Reply: We used 1Fe unit cell and unfolded Brillouin zone throughout the paper (except in Fig. S5). We have clarified this in the beginning of the paper (line 7~8 of the abstract).

Page 2:

Comment 1: Great, this is the info that I was looking for. Please move it earlier in the paper.

Reply: We have added specification in the beginning of the paper (line 7~8 of the abstract).

Page 3 - 6: Comments are addressed above.

Page 7:

Comment: New sentence.

Reply: Yes here should be the beginning of a new sentence. We have corrected it.

Page 13 - 14: Comments are addressed above.

Page 15:

Comment: What energy is the inset acquired at? Also the inset in panel d on surface B?

Reply: The inset of Fig. 3a (type A surface) is taken at $E=1.2$ meV, and the inset of Fig. 3d (type A surface) is taken at $E=1.7$ meV. The mapping energies are labelled in these images.

Page 16:

Comment 1: Is this a single atom Rb vacancy? So it's different from the larger pit that was shown in Fig 1c?

Reply: Yes this should be a single Rb vacancy. The vacancy shown in Fig. 1c is larger so we can image the lattice inside of it.

Comment 2: Please state the setup conditions for these spectra.

Reply: The setup condition is $V_b = 1.8\text{mV}$, $I = 100\text{pA}$ and $\Delta V = 50\mu\text{V}$ for all the spectra shown in Fig. 4b.

Page 17:

Comment 1: These topos are really hard to interpret because they haven't been leveled properly. You should do plane subtract on a single terrace in each topo (and use the same plane for the whole image), instead of fitting a single plane to the whole 2-terrace topo.

Reply: As the reviewer suggested, in the revised manuscript we have leveled them with a plane fitted by a single terrace.

Comment 2 and 3: addressed above

Page 19: Comments are addressed above.

Reviewer #2 (Remarks to the Author):

In this MS the authors show their STM results, which, they argue, indicates the existence of the novel nematic order in strongly hole-doped RbFe_2As_2 . Interestingly, they found that the nematic order breaks the symmetry between (π, π) and $(\pi, -\pi)$ directions rather than between $(\pi, 0)$ and $(0, \pi)$ direction, as in other Fe-based materials. They also argue that the superconducting gap is nodal and possibly has dx^2-y^2 symmetry. The results are interesting and worth publishing. However, I am not positive that they meet Nat. Comm. criteria by the following reasons:

1. This is not the first report of possible diagonal nematic order in strongly hole-doped Fe-pnictides. Earlier evidence was presented in Ref. [63] back in 2016.

We thank reviewer 2 for recognizing the novelty of our work. We think although it may not be the first report, our work is still of enough significance, since that:

1). We used a very different technique (STM) and provided more direct visualization of the diagonal nematicity, as well as the evidence of d-wave pairing component. To our knowledge, this is the first QPI measurement on the electron structure of AFe_2As_2 ($\text{A}=\text{K}, \text{Rb}, \text{Cs}$) and the first phase-sensitive STS study on the pairing symmetry of AFe_2As_2 .

2). In this revision, we performed further experiment on the surface doping effect of RbFe_2As_2 (Fig. 6). We found the diagonal nematicity is suppressed upon electron doping and the superconductivity is significantly enhanced, suggesting an anti-correlation between them. We also found the diagonal nematicity persists into the normal state above T_c , which implies the superconductivity forms out of this state and nematic fluctuations may play a role. These results give important clues on the origin of diagonal nematicity and highlight its close relation with superconducting pairing.

Such meaningful information is not provided by Ref. 63.

3). We note that the NMR result in Ref. 63 (arXiv:1611.04694(2016)) has not been published in any peer-reviewed journal.

2. *The diagonal nematic order should give rise to elongation of the hole pocket along either (p,p) or (p,-p) direction. The authors supposedly extracted the dispersion from STM, but didn't show the actual shape of the hole pocket. For comparison, STM study FeSe demonstrated that a hole pocket is elongated along Y direction [Sprau et al, Science 357, 6346 (2017)], consistent with axial nematic order.*

Yes the anisotropic QPI we observed should arise from anisotropic band structure. However, since QPI only measures the scattering vectors, it is usually difficult to derive the original Fermi surface solely from QPI, particularly when the Fermi surface has strong anisotropic structures or multiple pockets. In Sprau *et al.*'s paper, they used a *calculated* Fermi surface (which already has twofold symmetry) to interpret their QPI data, thus the dominating scattering vectors were known in prior (as shown in their Fig. 1). In our case, the QPI in the nematic state displays complicated behavior. For type A surfaces (commonly observed surface), the pattern is elongated along $(\pi, -\pi)$ at $E > 0$ but switched to (π, π) at $E < 0$. Such a behavior should arise from scatterings in between distorted band(s) (refs. 49-51), but one will need detailed knowledge of band structure to interpret them. However, band calculations with considering possible magnetic/orbital orders of RbFe_2As_2 have not been done before. We noticed these calculations are very sensitive to material parameters such as Hund's coupling strength and on-site Coulomb interactions (refs. 49-50), which are not easy to determine for this strongly correlated system. Therefore, at this stage a quantitative interpretation of the anisotropic QPI is beyond the scope of this paper.

On the other hand, in revised manuscript we present new results on surface K doped RbFe_2As_2 (Fig. 6). K atoms lowered the hole-doping of the surface FeAs layer. We found that the QPI pattern changed to be fourfold symmetric after K dosing, indicating the diagonal nematicity is suppressed. In such a roughly C_4 state, the QPI can be attributed to intra-band scattering of a more- C_4 symmetric hole band.

3. *The authors wrote in the abstract "Our results highlight the intimate connection between nematicity and superconducting pairing". I didn't find this connection. The authors discuss possible d-wave gap symmetry, but its relation to diagonal nematicity (if any) is unclear.*

To further illustrate this connection, we performed additional experiments and gained more information on the connection between diagonal nematicity and superconductivity. Based on all the data, such a connection is reflected in the following aspects:

1). In the phenomenal aspect, the diagonal nematicity is suppressed upon electron doping and the superconductivity is subsequently enhanced, which suggests an anti-correlation between them. This is analogous to the lightly doped FeSCs in which an

anti-correlation between $(\pi, 0)$ nematicity and superconductivity is also observed. Moreover, the diagonal nematicity is found to persist into the normal state above T_c . These observations imply the superconductivity in RbFe_2As_2 emerges out of the diagonal nematic phase and coexist with it.

2). In the mechanism aspect, it is known the $(\pi, 0)$ nematicity of lightly doped FeSCs is closely related to $(\pi, 0)$ stripe AFM order; while in optimal doped FeSC the $(\pi, 0)$ nematicity is substantially weakened and the electron pairing is believed to be s_{\pm} wave and mediated by $(\pi, 0)$ spin fluctuations. Here in the over-hole-doped regime, the re-emergence of a new type of (π, π) nematicity and change in the pairing symmetry with suppressed T_c imply the system approaches a new regime with different fluctuations. As theoretically expected, an AFM Mott insulating state at $3d^5$ configuration is likely the candidate. The related spin/nematic fluctuations may give rise to or mediate both the diagonal electronic nematicity and the d-wave pairing component, although the microscopic details certainly require further theoretical study.

A few additional remarks:

1. On p. 2, the authors related d-wave gap symmetry with (p,p) spin fluctuations. How this is related to RbFe_2As_2 in which neutron scattering and NMR data (see, e.g. Ref. [64]) show that spin fluctuations are nearly stripe-type (i.e., the peak is at $(p(1+\delta), 0)$ and related points?

We agree that the (π, π) spin fluctuations have not been directly observed in neutron scattering and NMR studies. However, we noticed that so far direct neutron scattering measurement on RbFe_2As_2 is still lacking (previous results are mainly on KFe_2As_2), and the NMR data without k-space resolution may not completely exclude other types of spin fluctuations aside of the $(\pi(1\pm\delta), 0)$ spin fluctuations. Our data also does not exclude the s-wave pairing component which may coexist with the d-wave pairing, as discussed in page 8, paragraph 2. Since the physics in such a strong correlated system has not been settled down, our results provide new evidences that pointing to the role of possible other fluctuations, which could be either spin or orbital fluctuations likely around (π, π) . We certainly agree that their existence should deserve further refined study. As ref. 64 does not directly support our conclusions, we have revised the statement on citing this paper (page 10, paragraph 2).

2. What is "stripe-type AFM (p,p) order" (p. 5 of the MS), and how it can promote diagonal nematicity?

Here we mean the RbFe_2As_2 may be proximate to a "stripe-type" AFM with Q along (π, π) direction, which could be analogous to the "double stripe" AFM order of FeTe with $Q = (\pi/2, \pi/2)$. This is because stripe-type AFM naturally breaks the C_4 rotation symmetry. It has been shown that the common $(\pi, 0)$ stripe AFM tends to open a partial gap along the antiferromagnetic direction (refs. 49-51), which result in a distorted Fermi surface and introduce $(\pi, 0)/(0, \pi)$ symmetry breaking in QPI. Analogously, our data may suggest there could be $(\pi, \pi)/(\pi, -\pi)$ fluctuations, which

introduced $(\pi, \pi)/(\pi, -\pi)$ symmetry breaking in QPI.

Certainly, at this stage we don't know whether such (π, π) stripe-type AFM do exist and its specific form. In the revised manuscript, we have revised this part with more cautious statements (Page 6, paragraph 2, line 12~24).

More generally, the authors wrote in the discussion that (p,p) spin fluctuations are likely candidate for promoting a diagonal nematicity. How this is possible if, as the authors stated, only one hole pocket at Γ has been observed in 2Fe zone?

The nematic order can be driven by Fermi surface instability or local interactions such as commensurate orbital ordering. For the former case, the QPI pattern may suggest there is a single hole pocket at Γ , however this does not mean there is no other pocket at different locations of BZ. Since STM is more sensitive to the band with large tunneling probability, the Γ band with small k (which have large tunneling probability) usually dominates the QPI. We notice that in ref. 37, ARPES measurement did observe two hole-like bands at the M' (π, π) point of 1Fe BZ. They didn't show up at Γ due to the weak folding potential or enhanced two-dimensionality of RbFe_2As_2 . These pockets can give rise to spin fluctuations along (π, π) through the Γ - M' scattering, as suggested by FRG calculations in ref. 22. On the other hand, it may also be possible even with one pocket. For example, cuprates usually have one Fermi surface, while it can also support (π, π) spin fluctuations.

For the latter case, as shown in FeSe and $\text{FeTe}_{1-x}\text{Se}_x$ (refs. 7, 15, 53), local interactions or orbital ordering may play a critical role and introduce symmetry breaking in QPI. Related discussions can be found in page 6, paragraph 2, line 24~28 of the revised manuscript, where we also clarify these two cases. Currently, it is not clear what is the origin of our experimental findings.

Reviewer #3 (Remarks to the Author):

The manuscript of Liu et al present a study of superconductivity and the electronic structure in RbFe_2As_2 by low temperature scanning tunnelling microscopy and spectroscopy and quasi-particle interference imaging. The results are novel and interesting, and in principle suitable for publication in Nature Communications. There are a few issues the authors should address however prior to publication.

The authors claim that they have discovered a novel type of nematicity which is rotated by 45deg compared to that of other iron pnictides, though similar to the electronic anisotropy seen in FeTe . The interpretation hinges on the assignment of the atomic structure of the surface: the authors assign the atomic resolution to a $\sqrt{2} \times \sqrt{2}$ reconstruction of the surface, in which case the symmetry breaking differs indeed by 45deg from that of other pnictides. If, however, one would assign the atomic resolution to either a complete Rb layer or the As layer, the electronic anisotropy would be the same as in other pnictides. Because the interpretation hinges on the assignment of the

atomic resolution, this point deserves further clarification by the authors:

- is there any other evidence (apart from the images shown in the manuscript) that the surface exhibits a $\sqrt{2}\times\sqrt{2}$ reconstruction? (e.g. LEED? Cleaves which result in different terminations?)*

We thank reviewer 3 for pointing out the significance of our work and recommendation of publication. We fully understand reviewer's concern that determining the orientation of the surface Rb lattice (with respect to the bulk FeAs lattice) is crucial for this work. In this revision, we further confirmed the orientation of surface $\sqrt{2}\times\sqrt{2}$ lattice by combining Laue diffraction measurement and STM imaging, and the results are summarized in Fig. S5. We first determined the orientation of \mathbf{a}_0 and \mathbf{b}_0 (the in-plane basic vectors of 2Fe unit cell) of RbFe_2As_2 single crystal by comparing its measured Laue pattern with the simulated pattern (Figs. S5a and S5b, see supplementary materials section III for details). Then the crystal was glued on STM sample holder with \mathbf{a}_0 and \mathbf{b}_0 aligned to X and Y scan directions, respectively. The STM image of cleaved surface (Fig. S5c) then directly shows the surface $\sqrt{2}\times\sqrt{2}$ lattice is rotated 45° with respect to \mathbf{a}_0 and \mathbf{b}_0 . Moreover, to ensure such method is reliable, we repeated the same procedure on a pure FeSe single crystal, the results are shown in Fig. S5d-f. FeSe displays a Laue pattern analogous to RbFe_2As_2 , its \mathbf{a}_0 , \mathbf{b}_0 directions are determined in similar way. STM image show that its surface lattice has a constant of 3.75 \AA with basic vectors the same as \mathbf{a}_0 , \mathbf{b}_0 . This is well expected for a Se terminated surface of pure FeSe. Therefore, the above measurement confirmed the $\sqrt{2}\times\sqrt{2}$ ($R45^\circ$) reconstruction on surface of RbFe_2As_2 .

- If the Rb reconstructs into a $\sqrt{2}\times\sqrt{2}$ order, this must mean that diffusion takes place between cleavage and imaging. Have the authors tried to cleave at lower temperatures to prevent diffusion from occurring?*

As reviewer suggested, in this revision we cleaved the sample at lower temperature ($\sim 30\text{K}$). This is achieved in another STM system where the sample (with a cleaving post) can be pre-cooled with liquid helium (Note for the millikelvin STM the lowest cleaving temperature we can reach is $\sim 80\text{K}$). As shown in Fig. S4a, we indeed observed domain structures at some regions of low-T cleaved sample. Fig. S4b is an atomically resolved image near a domain boundary. Aside of the boundary, the surface still shows a $\sqrt{2}\times\sqrt{2}$ lattice with a constant of 5.4 \AA . However, the lattice of the upper domain is found to be shifted by $1/2$ constant (along both \mathbf{a} and \mathbf{b} directions) with respect to lower domain, as illustrated by the lattice of white spots (see figure caption). The existence of different domains, as the reviewer had pointed out, is well expected from the surface lattice model sketched in Fig. 1d, which gives a further support of this model. Note that previously we didn't observe domain boundaries at the cleaving temperature of 80K , which is possibly because the domain size is too large at such temperature. At lowered temperature the diffusion rate is reduced so the domain size is also limited. To completely prevent surface diffusion, one would need even lower temperature to cleave, which we cannot achieve at this stage. Related

discussion on domain structure is added in revised main text (page 3, paragraph 2, line 10-13) and supplementary materials (page 2, last paragraph).

• *there should be domains of $\sqrt{2}\times\sqrt{2}$ order. Have the authors seen evidence of these domains?*

This is a very good point. As discussed above, we do observe domain structures on samples cleaved at lower temperature (Fig. S4). This further supports our surface lattice model shown in Fig. 1d.

The atomic distances are not a very good indicator for the surface termination, as the calibration will change with temperature (which can be particularly an issue in a millikelvin system) – unless the authors have calibrated their instrument independently at the same temperature as the topographic images shown, this can yield misleading assignments.

We understand reviewer's concern. The data shown in the manuscript, except the surface K-dosing data in this revision, are all taken at the lowest temperature (20mK) in a millikelvin STM. Previously we have calibrated the piezo scanner at the same temperature by other materials, such as HOPG and Bi_2Se_3 (Cu doped). The measured lattice constants are all consistent with reported values (with an error $< 0.05\text{\AA}$), as shown by the atomically resolved images below (Fig. R3). Therefore we are confident with our results on RbFe_2As_2 . Moreover, in this revision we measured the RbFe_2As_2 in a different STM system at $T=4.5\text{K}$, and we still observed the same $\sqrt{2}\times\sqrt{2}$ lattice with a constant of 5.4\AA , which further confirms our conclusion.

Figure R3 | (a) Atomically resolved STM image of HOPG ($3.5\times 3.5\text{ nm}^2$, $V_b = 80\text{mV}$, $I = 0.5\text{nA}$), taken at $T=20\text{ mK}$. The lattice constant is measured to be $a=b= 2.50\text{\AA}$, consistent with the expected value (2.47\AA). (b) Atomically resolved STM image of $\text{Cu}_x\text{Bi}_2\text{Se}_3$ ($7\times 7\text{ nm}^2$, $V_b = 20\text{mV}$, $I = 20\text{pA}$), taken at $T=20\text{ mK}$. The lattice constant is measured to be $a=b= 4.14\text{\AA}$, consistent with previously reported value of 4.1\AA (Levy *et al.*, **PRL** 110, 117001 (2013)).

Minor issues:

• *There is an issue with the Dynes function, eq. 1 – the modulus should be outside of*

“Re”.

Yes this is a mistake and we have corrected it in the revised manuscript. We thank reviewer for pointing it out. We have checked that the original function used in the DOS calculation was correct.

- *What is the setpoint condition used for the maps in fig. 2? Is it the same in both cases? What lock-in modulation was used for the maps?*

In Fig. 2 (and Figs. S7, S8), all the dI/dV maps were taken at a bias voltage (V_b) equal to the mapping energy (labeled on each map) and $I = 100\text{pA}$. To obtain good signal to noise ratio, the amplitude of lock-in modulation (ΔV) for each map was adjusted to be 5% of V_b . In Fig. 6 and Fig. S10, all the dI/dV maps were taken at the same setpoint of $V_b = 10\text{mV}$, $I = 200\text{pA}$, $\Delta V = 1\text{mV}$.

- *Is it clear that figs. 1a and b are different terminations? What is the evidence for this? If it is spectroscopic, could it just be different tips sensitive to different orbitals? (as seen, e.g., in FeSe, see Science 357, 75). From the crystal structure, there doesn't seem to be an obvious way to have two different surface terminations which both have the same $\sqrt{2}\times\sqrt{2}$ superstructure of Rb.*

The surface shown Figs. 1a and b have the same lattice structure and very similar tunneling spectrum in large energy scale (Fig. 1e). Therefore they should be the same terminations. Their only difference is the strength of anisotropy in the QPI pattern observed near E_F . The STM tips we used (made by Pt) were all pre-cleaned by e-beam heater and were carefully treated on Au(111) to ensure they are metallic with featureless DOS near E_F . Therefore we do not think the difference came from the tip. We think it could be due to local strain, which enhanced the nematicity in type B surfaces.

- *Why does the resistance drop below zero? (fig. S1c)*

This is due to measurement noise. The low-temperature resistance of RbFe_2As_2 sample is quite low ($\sim 10^{-7}$ ohm) so the noise became relatively large. In the revised Fig. S1c we show both raw R-T data and the smoothed R-T curve.

- *Pg. 4: re the statement that “Such C_4 symmetry breaking between (π, π) and $(\pi, -\pi)$ has never been reported for other iron pnictides.” This is true for pnictides, but not for chalcogenides (i.e. FeTe). This is briefly discussed in the discussion section, but could be mentioned earlier in the introduction.*

We have added related discussion on FeTe in the introduction. (Page 1, line 6-7 of introduction)

- *In the discussion of the relation between nematicity and superconductivity, the*

authors may also want to compare their work to Sci. Adv. 1, e1500206, which has investigated this relation for FeSeTe.

Yes this is a related STM paper on nematicity. We have cited this paper (ref. 53 of the revised manuscript) and added related discussions. (Page 6, paragraph 2, line 4 from bottom; Page 7, line 12)

- *Fig. 1: please provide temperature at which this has been taken*

The data shown in Fig. 1 (and also Figs. 2 - 5) are all taken at the base temperature of 20 mK.

- *Caption of fig. 2, please indicate whether images are symmetrized (this is clear from the supplementary, but should be also clearly stated in the caption in the main manuscript)*

Yes the FFT images in Fig.2 are symmetrized. We have added statement on this in Fig. 2 caption of revised manuscript.

- *Please provide setpoint conditions for all topographic images and setpoint condition and lock in parameters for all spectroscopic maps and spectra (also in supplementary material). Also temperatures of the STM should be indicated for all measurements (not just a noise temperature), unless all have been taken at 20mK (in which case this should be indicated as well).*

We have added the setpoint conditions, lock-in parameters for all the topographic images and tunneling spectra/dI/dV maps in the manuscript. The data shown in Figs. 1 – 5 are taken at 20 mK, and the data shown in Fig. 6 are taken at 4.5K. We have specified the measurement temperatures in all the figure captions in the manuscript.

Reviewers' comments:

Reviewer #1 (Remarks to the Author):

I still have a number of concerns about this manuscript:

(1) The overall message isn't clearly communicated. If the authors are trying to say something about the phase diagram, they need to show a phase diagram. If the authors are trying to say something about the relationship between rotational-symmetry-breaking and band structure (as measured by QPI), then they need schematics of the band structure and how the symmetry-breaking affects it. As it is, the paper reads like a long, detailed list of observations.

(2) The authors show in Fig S2 that the base electron temperature of the sample in their measurements is 310 mK. This is the temperature that matters. It is irrelevant if some thermometer elsewhere on their dilution fridge reads 20 mK. The authors should eradicate all mentions of 20 mK from their manuscript.

(3) The authors acknowledge in their rebuttal letter that (π, π) symmetry breaking has been reported (albeit not published) in another iron-pnictide CsFe_2As_2 (arxiv 1611.04694). Then their statement on page 4 of the manuscript is false: "Such C_4 symmetry breaking between (π, π) and $(\pi, -\pi)$ has never been reported for other iron pnictides."

(4) I'm concerned that all of the QPI q -space data in this paper is symmetrized. If the material is indeed breaking the symmetry along (π, π) but not $(\pi, 0)$ then symmetrization would be appropriate. But if all we ever see is the post-symmetrized data, then we can't judge whether the symmetry axis was appropriately chosen. We need to see the raw q -space data.

(5) Figs 4 & S9: 4 points are not sufficient to fit an exponential decay. In Fig 4, there is obviously additional inhomogeneity near the defect in question, so the background value is not well-established. In both Figs 4 & S9, the 4 points are not even monotonically decreasing, so there are really only 3 valid points. The exponential decay is $A \cdot \exp(-B \cdot r) + C$, so it's a 3-parameter fit to 3 points. This figure does not belong in the main text, and cannot be reasonably used to justify any important argument in the manuscript.

(6) Fig 6j,k: these figures show that K -doping, which the authors assume to electron-dope the material, actually moves the hole pocket up, i.e. adds more holes! This is a serious problem with the interpretation of the results. The authors acknowledge this problem in the last sentence of the results section, but they hand-wave it away, without giving any concrete proposal for what kind of band reconstruction could lead to this bizarre apparent contradiction.

(7) Rebuttal page 2: The authors claim that they can't detect the strain via the spatial lockin technique proposed by Lawler, Nature 2010 (10.1038/nature09169) because they don't have dI/dV maps. But the authors misunderstand Lawler's algorithm. It doesn't require dI/dV . It can operate on the topography. If the authors can use a single tip to image both regions A & B, i.e. with identical xy piezo calibration, then the use of Lawler's algorithm can detect tiny variations in strain between different regions. For example, see Gao, PNAS 2018 (10.1073/pnas.1718931115). I would like to see the authors implement this algorithm, rather than guess about the relative strain in regions A vs. B.

(8) On all E -vs- q plots (e.g. Fig 2e,i,j) authors should superimpose the white dashed guide to the eye only on half of the plot, so that we can see the unobscured data on the other half.

(9) Authors should clarify in all captions whether bias modulation was rms or peak-to-peak.

Reviewer #2 (Remarks to the Author):

I read the revised version of the manuscript and the authors reply to my comments and to the comments by the other two referees.

Overall, my opinion is that they authors presented honest description of what their results are, what they understand, and what they do not understand. The origin of "diagonal" nematicity is totally unknown, and I am pleased that the authors softened the statements at the beginning of the paper. I am also pleased that they do not state as "the fact" that SC in RbFe₂As₂ is d-wave, as there are arguments for both d-wave and s-wave gap symmetry is strongly hole-doped Fe-pnictides.

As about "novelty", it is true that the first reports about possible diagonal nematicity came from 2016 NMR paper. But I agree with the authors that STM provides more direct evidence of a nematic order.

Reviewer #3 (Remarks to the Author):

The authors have addressed all the points I raised in the revised manuscript. It is good to see that they have made a real effort to convince the referees and provide more supporting data where possible. I recommend the manuscript for publication.

Reply to reviewers:

Our point-by-point responses to all the comments are in blue text below and the original comments are in *italic*. New revisions are highlighted in green color in the manuscript.

Reviewer #1 (Remarks to the Author):

(1) The overall message isn't clearly communicated. If the authors are trying to say something about the phase diagram, they need to show a phase diagram. If the authors are trying to say something about the relationship between rotational-symmetry-breaking and band structure (as measured by QPI), then they need schematics of the band structure and how the symmetry-breaking affects it. As it is, the paper reads like a long, detailed list of observations.

Our results indicate a new nematic phase emerges in the deep hole-doping region of iron pnictides, which co-exists with a nodal $d_{x^2-y^2}$ like pairing. We agree a global phase diagram would help readers to understand the results more easily. In this revision we have added a sketch of phase diagram in Fig. 7 (and is also shown below). It is known that by doping holes to parent compound, the $(\pi, 0)$ stripe AFM and nematicity weakened and disappeared near optimal doping. The pairing at optimal doping is believed to be nodeless s_{\pm} wave and mediated by $(\pi, 0)$ spin fluctuations. Here in the over-hole-doped regime, the re-emergence of a new type of diagonal nematic state and change in the pairing symmetry imply the system approaches a new regime with different fluctuations. Since an AFM Mott insulating state was predicted for $3d^5$ configuration, the related spin/nematic fluctuations are likely the candidate. Therefore our results give important clues on the origin of nematicity, and highlight the intimate relation between nematicity and superconducting pairing.

Fig. | Schematic phase diagram of 122 type iron-pnictides. The blue area denotes $(\pi, 0)$ stripe AFM

that coexists with $(\pi, 0)$ nematicity, and the light blue area denotes $(\pi, 0)$ nematic fluctuation. The two green areas are superconducting dome. The red area denotes the discovered (π, π) nematicity at the $3d^{5.5}$ configuration (which could become (π, π) nematic fluctuation at increased temperature). For the hole-doping superconducting dome, electron pairing is nodeless (s_{\pm}) in the middle of the dome, but became nodal ($d_{x^2-y^2}$ like) as approaching $3d^{5.5}$. A Mott insulator phase is theoretically expected at the $3d^5$ configuration. Upper insets demonstrate the rotational symmetry breaking in $(\pi, 0)$ and (π, π) nematic state.

(2) The authors show in Fig S2 that the base electron temperature of the sample in their measurements is 310 mK. This is the temperature that matters. It is irrelevant if some thermometer elsewhere on their dilution fridge reads 20 mK. The authors should eradicate all mentions of 20 mK from their manuscript.

The effective electron temperature (T_{eff}) describes the energy resolution of the system, which can be affected by external noise such as RF radiation. It is true that effective electron temperature is more important for STS measurement (that is why we calibrated it carefully). However, the actual or say lattice temperature of the system is also essential. For example the piezo calibration highly depends on it, as emphasized by Review 3's comment previously. The scanner of our millikelvin system is calibrated at $T=20\text{mK}$ and all the experiment in this system are performed at $T=20\text{mK}$. We don't think we should remove the description of this condition. We have clearly specified these two different temperature at the beginning of the experimental part (page 2, last paragraph) and distinguished them by different symbols (T_{eff} and T) in our original manuscript. In this revision we have added $T_{\text{eff}} = 310\text{mK}$ on every place we mentioned the temperature of the millikelvin system.

(3) The authors acknowledge in their rebuttal letter that (π, π) symmetry breaking has been reported (albeit not published) in another iron-pnictide CsFe_2As_2 (arxiv 1611.04694). Then their statement on page 4 of the manuscript is false: "Such C_4 symmetry breaking between (π, π) and $(\pi, -\pi)$ has never been reported for other iron pnictides."

This is a kind of misunderstanding. This statement is in the context where we are specifically discussing the QPI measurement of RbFe_2As_2 . What we mean is that *in QPI measurement*, the C_4 symmetry breaking between (π, π) and $(\pi, -\pi)$ has never been reported for other iron pnictides, which is true. We have clarified this point in the revised manuscript (page 4, paragraph 2, line 11).

(4) I'm concerned that all of the QPI q -space data in this paper is symmetrized. If the material is indeed breaking the symmetry along (π, π) but not $(\pi, 0)$ then symmetrization would be appropriate. But if all we ever see is the post-symmetrized data, then we can't judge whether the symmetry axis was appropriately chosen. We need to see the raw q -space data.

The raw QPI and q-space FFT data were already presented in Fig. S7, S8 and S10 of the previous supplementary materials, which clearly show the two-fold symmetry breaking along (π, π) . We have also specified the symmetrization process in detail according to reviewer's own comments in the last review. If reviewer means this is still not clear enough for some readers, and in this revision we have put part of the raw FFT data into the main text (in Fig. 2). We emphasize that the raw data cannot be put entirely into the main text due to limited space, however they are certainly provided. We note that in the previous review, reviewer 1 commented: "*Regarding claim (1): The data clearly justifies this claim. As far as I know, the 45deg nematicity is unique...*". The claim (1) is: "*They use quasiparticle interference imaging, and vortex core state imaging, to show an unusual nematicity (rotational symmetry breaking) along the diagonal direction...*". Thus we would assume reviewer 1 had checked our QPI data carefully before making this comment.

(5) Figs 4 & S9: 4 points are not sufficient to fit an exponential decay. In Fig 4, there is obviously additional inhomogeneity near the defect in question, so the background value is not well-established. In both Figs 4 & S9, the 4 points are not even monotonically decreasing, so there are really only 3 valid points. The exponential decay is $A \cdot \exp(-B \cdot r) + C$, so it's a 3-parameter fit to 3 points. This figure does not belong in the main text, and cannot be reasonably used to justify any important argument in the manuscript.

We don't see a reason for why 4 points are not sufficient to fit an exponential decay. We used a standard least square method to fit the data and the fitting converges, so the fitting parameters are reliable. A fitting should not depend on whether the data points are monotonic or not, as there is always noise in the experiment. All the data points in Fig. 4 and S9 are valid experimental data (note the data points in Fig. S9 are clearly monotonic), and 4 points are capable of fitting an equation with 3 parameters. Although there are some small inhomogeneity in Fig. 4(a), the STS are taken at the clean areas but not on these inhomogeneity (tip position is marked). The STS in Fig. S9 are taken on a very homogeneous surface. In this revision we have moved Fig. S9 into the main text (in Fig. 4) so it will give stronger support to our conclusion, as suggested by the reviewer. Reviewer may mean the fitting may not be very accurate because there are not many points, however, our conclusion (about sign-changing pairing) does not sensitively rely on the exact value of decay length. As long as the gap suppression is observed on the defect site (and it is), it evidences sign-changing pairing. Moreover, our claim on d-wave pairing component is not only based on impurity effect, as we stated that it cannot distinguish d-wave and nodal s-wave pairing, much stronger evidence is from the effect of the step edges shown in Fig. 5.

(6) Fig 6j,k: these figures show that K-doping, which the authors assume to electron-dope the material, actually moves the hole pocket up, i.e. adds more holes! This is a serious problem with the interpretation of the results. The authors acknowledge this problem in the last sentence of the results section, but they hand-wave it away, without

giving any concrete proposal for what kind of band reconstruction could lead to this bizarre apparent contradiction.

The breakdown of rigid band picture upon doping is common for strong correlated systems. QPI is not a direct method to map out the band structure, it only measures the strong scattering vectors with relatively high tunneling probability. It is also difficult to predict band reconstruction induced by nematicity without knowing its mechanism (refs. 49-51), and the calculation usually strongly depends on parameters such as Hund's coupling and on-site Coulomb interaction, which are unknown at this stage. Our conclusion is mainly based on the observation of symmetry breaking. The key information in Fig. 6 is that the symmetry is restored to C4 after electron doping and superconductivity gap became larger. This is a new information pointing to the relation between (π, π) nematicity and superconductivity. Detailed evolution of the Fermi surface will require more specific measurement and calculations with considering correlation effect and possible spin/orbital orders, but they are beyond the scope of this work.

(7) Rebuttal page 2: The authors claim that they can't detect the strain via the spatial lockin technique proposed by Lawler, Nature 2010 (10.1038/nature09169) because they don't have dI/dV maps. But the authors misunderstand Lawler's algorithm. It doesn't require dI/dV. It can operate on the topography. If the authors can use a single tip to image both regions A & B, i.e. with identical xy piezo calibration, then the use of Lawler's algorithm can detect tiny variations in strain between different regions. For example, see Gao, PNAS 2018 (10.1073/pnas.1718931115). I would like to see the authors implement this algorithm, rather than guess about the relative strain in regions A vs. B.

The strain effect is only our speculation, whether the strain is detected or not, it does not affect our main conclusion that the (π, π) nematicity widely exists, although its strength could vary for different samples. We have stated in our previous reply that the chance of finding type B surfaces is small. Particularly, to distinguish surface A and B, we still need to do time-consuming dI/dV mapping to check the nematic state. Thus such measurement will take unpredictable long time and the tip state could easily change during long time searching and mapping, and may not give meaningful results. We agree characterizing strain would help to further understand the mechanism of nematicity, however it is still technically challenging for the current sample at this stage. The main finding of this paper is the observations of (π, π) nematicity and its relation/evolution with superconducting pairing, on which we have spent plenty of effort. Quantifying the strain deserves further study but is beyond the scope of this paper.

(8) On all E-vs-q plots (e.g. Fig 2e,i,j) authors should superimpose the white dashed guide to the eye only on half of the plot, so that we can see the unobscured data on the other half.

We have done this in Figs. 2f~h and Figs. 6j, k in the revised manuscript.

(9) *Authors should clarify in all captions whether bias modulation was rms or peak-to-peak.*

The symbol “ ΔV ” stands for the Amplitude of bias modulation (a cosine waveform). We have clarified this in the Method section before (page 9, green highlighted in Method section).

Reviewer #2 (Remarks to the Author):

Overall, my opinion is that they authors presented honest description of what their results are, what they understand, and what they do not understand. The origin of "diagonal" nematicity is totally unknown, and I am pleased that the authors softened the statements at the beginning of the paper. I am also pleased that they do not state as "the fact" that SC in RbFe₂As₂ is d-wave, as there are arguments for both d-wave and s-wave gap symmetry is strongly hole-doped Fe-pnictides.

As about "novelty", it is true that the first reports about possible diagonal nematicity came from 2016 NMR paper. But I agree with the authors that STM provides more direct evidence of a nematic order.

We thank Reviewer #2 for understanding our effort and pointing out the significance of our work, as STM is a more direct probe on the nematic electronic state.

Reviewer #3 (Remarks to the Author):

The authors have addressed all the points I raised in the revised manuscript. It is good to see that they have made a real effort to convince the referees and provide more supporting data where possible. I recommend the manuscript for publication.

We thank Reviewer #3 for the recommendation of publication and recognizing our effort on providing more supporting data where possible.

REVIEWERS' COMMENTS:

Reviewer #3 (Remarks to the Author):

Following the resubmission of the manuscript by Liu et al, referee 1 has raised in particular again the issue of determination of strain in the sample as well as whether the evidence for symmetry breaking from QPI is sufficient and representative of the raw data.

Re the determination of strain (point 7 by referee 1), while it would be very valuable to have this information, it is difficult to obtain from STM images and would need very careful calibration and, as the referee points out, imaging of areas of surface A and B within the same image. It is not clear from the data presented whether such areas exist, but also this is not central to the claim made by the authors.

Re the QPI (point 4 by referee 1), I am content that the authors provide unsymmetrized data which support their conclusions. Could surfaces type A and B be the result of two different tips probing predominantly electronic states of different orbital character (compare, e.g., Fig. S6 of Science 357, 75)?

I would disagree with referee 1 (point 2) that only the electronic (effective) temperature should be quoted, as that may well be influenced by noise originating from outside the cryostat. This is therefore a priori a rather technical point. As long as the electronic temperature is clearly stated this is sufficient.

It is good to see a phase diagram (fig. 7), however the authors may want to add some references (or reference to a review article) for where the information about the phases and phase boundaries comes from.

Overall, in my opinion the main claim is sufficiently justified and supported by the data to merit publication.

Reply to reviewer:

We thank Reviewer #3 very much for his/her comments regarding Reviewer #1's criticisms and the supporting of publication. Our responses are in blue text below, following the original comments. The corresponding revisions in the manuscript are highlighted.

Reviewer #3 (Remarks to the Author):

Following the resubmission of the manuscript by Liu et al, referee 1 has raised in particular again the issue of determination of strain in the sample as well as whether the evidence for symmetry breaking from QPI is sufficient and representative of the raw data.

Re the determination of strain (point 7 by referee 1), while it would be very valuable to have this information, it is difficult to obtain from STM images and would need very careful calibration and, as the referee points out, imaging of areas of surface A and B within the same image. It is not clear from the data presented whether such areas exist, but also this is not central to the claim made by the authors.

We did not observe an area in which surface A and B coexist. This is possibly because the domain size of surface A and B are much larger than the scan range of our STM ($\sim 0.5 \times 0.5 \mu\text{m}^2$), which make it difficult to find the boundary between them.

Re the QPI (point 4 by referee 1), I am content that the authors provide unsymmetrized data which support their conclusions. Could surfaces type A and B be the result of two different tips probing predominantly electronic states of different orbital character (compare, e.g., Fig. S6 of Science 357, 75)?

We agree that different tip states may detect different orbitals of the sample, as demonstrated by Sprau *et al.* in *Science* 357, 75(2017). However, here we did not observe notable change of the tip during QPI imaging, and the tips we used were all pre-cleaned by e-beam heating and carefully treated on Au(111) before conducting measurement on RbFe_2As_2 . Therefore we think the differences between surface A and B are likely not from the tip, but from the sample themselves (which could be due to local strain).

I would disagree with referee 1 (point 2) that only the electronic (effective) temperature should be quoted, as that may well be influenced by noise originating from outside the cryostat. This is therefore a priori a rather technical point. As long as the electronic temperature is clearly stated this is sufficient.

We fully agree with reviewer 3 on this point. We have clearly stated the effective electronic temperature throughout the manuscript.

It is good to see a phase diagram (fig. 7), however the authors may want to add some

references (or reference to a review article) for where the information about the phases and phase boundaries comes from.

According to reviewer's suggestion, we have added references for the phase diagram in figure 7's caption (highlighted text) and in the main text (page 9, paragraph 1, highlighted). We added one more review article (Ref. 2) about the phase diagram of FeSC in the reference list, and the reference index throughout the manuscript are corrected correspondingly.

Overall, in my opinion the main claim is sufficiently justified and supported by the data to merit publication.

We thank reviewer 3 again for the supporting of publication.